



# Opportunities and limitations related to the application of
# plant-derived lipid molecular proxies in soil science
**Boris Jansen[1] and Guido L. B. Wiesenberg[2]**
[1]{Institute for Biodiversity and Ecosystem Dynamics, University of Amsterdam,
Amsterdam, P.O. Box 94240, NL-1090GE, The Netherlands}
[2]{Department of Geography, University of Zürich, Winterthurerstrasse 190, CH-8057
Zürich, Switzerland}
*Correspondence to:* B. Jansen (B.Jansen@uva.nl)
## Abstract
The application of lipids in soils as molecular proxies, also often referred to as biomarkers,
has dramatically increased in the last decades. Applications range from inferring changes in
past vegetation composition, climate and/or human presence to unraveling input and turnover
of soil organic matter (SOM). Molecules used include extractable and ester-bound lipids as
well as their carbon or hydrogen isotopic composition. While holding great promise, the
application of soil lipids as molecular proxies comes with several constraining factors the
most important of which are: i) variability in the molecular composition of plant-derived
organic matter plant-internally and in between plant individuals; ii) variability in (relative
contribution of) input pathways into the soil; and iii) transformation and/or (selective)
degradation of (some of) the molecules once present in the soil. Unfortunately, the
information about such constraining factors and their impact on the applicability of molecular
proxies is fragmented and scattered. The purpose of this study is to provide a critical review
of the current state of knowledge with respect to the applicability of molecular proxies in soil
science, specifically focusing on the factors constraining such applicability. Variability in
genetic, ontogenetic and environmental factors influence plant *n*-alkane patterns in the way
that no unique compounds or specific molecular proxies pointing to e.g. plant-community
differences or environmental influences, exist. Other components such as *n*-alcohols, *n*-fatty
acids, cutin- and suberin-derived monomers have received far less attention in this respect.
Furthermore, there is a high diversity of input pathways offering both opportunities and





limitations for the use of molecular proxies at the same time. New modelling approaches
might offer a possibility to unravel such mixed input signals. Finally, transformation and
turnover of SOM offer opportunities when tracing such processes is the purpose of applying a
molecular proxy, whilst posing limitations when they obliterate molecular proxy signals
linked to other phenomena. For *n*-alkanes several modelling approaches have recently been
developed to compensate for (selective) degradation. Still such techniques are in their infancy
and information about their applicability to other classes of components than *n*-alkanes is
lacking yet. All constraining factors considered can have a significant influence on the
applicability of molecular proxies in soil science. The degree of influence strongly depends
on the type of molecular proxy as well as the environmental context in which it is applied.
However, the potential impact of the constraining factors should always explicitly be
addressed whenever molecular proxies are applied in a soil scientific context. More
importantly, there is still a serious lack of available information in particular for compound
classes other than the *n*-alkanes. Therefore, we urgently call for the consideration of more
holistic approaches determining various parameters during sampling as well as using as many
compound classes as possible.
**1    Introduction**
Since more than a century, various compounds deriving from the substance class of lipids,
which are operationally defined as soluble in organic solvents, but not or to a limited degree
in water, have been investigated in plant and soil science. Some of the earliest publications in
plant science date back to the first half of the 19[th] century (Liebig et al., 1837; Wöhler F. and
Liebig, 1839) and in soil science to the early 20[th] century as already reviewed by Stevenson
(1966). One of the main interests to study lipids apart from the general understanding of the
human diet was the large heterogeneity of compounds included in this substance class. Some
of the individual compounds have been described as 'biomarkers' or 'biogenic markers', i.e.
compounds that *"may be diagnostic of specific organisms, classes of organism, or general*
*biota that contribute organic matter to the atmosphere, aqueous or sedimentary*
*environment"* (Peters et al., 2005). In addition to these contemporary biogenic markers, also
referred to as 'geochemical fossils' (Tissot and Welte, 1984), in environmental sciences also
anthropogenic markers and petroleum markers were highlighted by Peters et al. (2005) that
have the ability to be preserved with "no or only minor change" (Tissot and Welte, 1984).
Eganhouse (1997) summarized the principal criteria for a specific marker as follows:



*"Molecular markers must be typical for specific sources and characterized by their*
*conservative behavior in environmental archives"*. In other disciplines such as medicine and
toxicology a variety of *"medical signs, symptoms, biomarkers, surrogate endpoints, clinical*
*endpoints, validation"* is used under the umbrella biomarker (Strimbu and Tavel, 2010).
Because *sensu strictu* the term biomarker has been used for the differentiation of biological
tissues of different origin in environmental matrices, during the recent years the term
'molecular proxy' has become more frequently used. This term allows for an inclusion of
biomarkers *sensu strictu* as individual compounds characterizing specific biogenic sources,
but also individual compounds acting as specific proxy e.g. for anthropogenic impact or
thermal alteration. Furthermore, it accommodates the use of groups of compounds used in the
before mentioned way. Finally, it implies the use of molecular ratios of compounds like the
carbon preference index (CPI) or the average chain length (ACL) that could also be indicative
for biogenic sources, alteration or overprint of organic matter. Therefore, in the present work
we use the term molecular proxy rather than biomarker.
In its broadest sense, molecular proxies allow determination of the presence, absence, or
certain characteristics of a (set of) molecule(s) that are indicative for a process in, or state or
composition of a system of interest. For instance, in the clinical sciences molecular proxies
among other applications are used as indicators of the presence of a disease or response to
treatment (Brennan et al., 2013; Van Bon et al., 2014); in toxicology to assess the effect of
toxicant exposure on biota (Clemente et al., 2014); in the forensic sciences to link suspects to
a crime scene (Concheri et al., 2011); in limnology to examine past lacustrine environmental
conditions (Castañeda and Schouten, 2011); and in organic geochemistry to follow oil
formation and translocation in source and reservoir rocks (Curiale, 2002).
Also in soil science, molecular proxies have been used for decades, and their application has
exponentially increased in the last decade as indicated by the number of related articles
published in Web of Science indexed journals (Table 1). Compared to the overall timeframe
covered by Scopus, between 23 % (pentacyclic triterpenoids) and 99 % (GDGTs = glycerol
dialkyl glycerol tetraethers) of the publications using molecular proxies in soil science have
been published in the last ten years (2006-2015). On average (± SEM) 59 ± 4 % of the
publications with the respective keyword selections have been published in the last decade.
This clearly illustrates a strong increase associated by a diversification of the use of
molecular proxies in soil science. The types of molecular proxies used are as diverse as the
field of soil science itself. They range from the use of phospholipid fatty acids to estimate



bacterial and fungal biomass in soils (Frostegard and Bååth, 1996), to the application of preserved retene/caldalene ratios to infer palaeoecological vegetation shifts (Hautevelle et al., 2006). Also the archives of the molecular proxies in soil sciences that are used are diverse and, in addition to soils themselves, include lacustrine and terrestrial sediments, peat deposits, as well as paleosols (Zhang et al., 2006; Bai et al., 2009; Andersson et al., 2011; Berke et al., 2012). However, in spite of this large variety a limited number of scientific topics can be discerned that encompass the great majority of molecular proxy application in the soil sciences. These are:

- Changes in vegetation composition inferred from extractable and/or ester-bound lipids of plant origin, and/or their carbon isotopic composition (e.g. Huang et al., 1996; Zech et al., 2009; Le Milbeau et al., 2013).

- Changes in climate, i.e. mean annual temperature and/or precipitation inferred from bacterial membrane lipids and/or the hydrogen isotopic composition of plant-derived lipids (e.g. Weijers et al., 2006; Krull et al., 2006; Rao et al., 2009).

- Changes in palaeoelevation inferred from bacterial membrane lipids and/or the hydrogen isotopic composition of plant-derived lipids (e.g. Sachse et al., 2006; Bai et al., 2011; Ernst et al., 2013).

- Changes in human impact or settlement inferred from compound-specific N isotope analysis or transformation products of plant-derived lipids, e.g. through burning, or manure derived lipids (e.g. Bull et al., 1999; Eckmeier and Wiesenberg, 2009; Zocatelli et al., 2012).

- Contribution of fossil fuel-derived carbon to soil assessed by lipid molecular composition and compound-specific isotopes (e.g. Lichtfouse et al., 1995; Lichtfouse et al., 1997; Rethemeyer et al., 2004).

- Input, transformation and/or decomposition of soil organic matter inferred from or traced through extractable and/or ester-bound lipids of plant origin and/or bacterial membrane lipids and/or their carbon isotopic composition. (e.g. Nierop et al., 2001; Amelung et al., 2008; Hamer et al., 2012).

In Table 1 an overview is given of the classes of molecules frequently used as molecular proxies in soil archives in relation to their application as well as total and recent (last ten years) publications including the respective keywords.

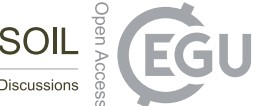

When using molecular proxies to answer research questions in any of the areas identified, in
particular when soils are used as an archive, several constraining factors have to be taken into
account that vary with the type of application and research question to be answered. The most
important ones are:
i)    Variability in the source of plant-derived organic matter, i.e. abundance and
6       composition of the molecular proxies in different plant species, plant specimens and
7       plant parts as a result of genetic or life stage variations and/or external factors such as
8       climate, seasonality or exposure to the sun (e.g. Nødskov Giese, 1975; Lockheart et
9       al., 1998; Shepherd and Griffiths, 2006).

ii)   Variability in (relative contribution of) input pathways into the soil, in particular
microbial versus vegetation input, and root versus aboveground biomass input (e.g.
Jackson et al., 1996; Schefuß et al., 2003; Mambelli et al., 2011).
iii)   Transformation and/or (selective) degradation of (some of) the compounds once
present in the soil, when it is not the aim of the study to use the molecular proxies to
study such transformations (e.g. De Leeuw and Baas, 1986; Nguyen Tu et al., 2004;
Andreetta et al., 2013).
However, the information about such constraining factors and their impact on the
applicability of molecular proxies is fragmented and scattered over different publications
inside and outside the scientific discipline of soil sciences. For instance, much of the
available information about variation of leaf wax lipid composition is presented in the plant
physiological literature in studies that were not conducted with the application of such lipids
as molecular proxy for past vegetation composition from soil archives in mind (e.g. Tulloch,
1973; Avato et al., 1984; Kim et al., 2007). The fragmentation of the information makes it
difficult for researchers to assess the potential influence of constraining factors on the
application of molecular proxies. It also hinders the identification of hiatuses in the available
knowledge about the constraining factors as well as the designation of potential strategies to
compensate or correct for such constraints.
Therefore, the purpose of the present study is to provide a critical review of the current state
of knowledge with respect to the applicability of molecular proxies in soil science,
specifically focusing on the factors constraining such applicability. Based on this we will
identify areas for future research both with respect to the application of molecular proxies in
soil science as well as the constraints thereof.





The vastness of the field of molecular proxies forced us to restrict the scope of the present
study. With respect to the molecules to consider, a first restriction was to focus on those
related to the earlier mentioned main areas of application of molecular proxies in soil science.
A second restriction was to focus on the main classes of components as used by several
researchers. Finally, in spite of their common application, we explicitly excluded lignin and
phospholipid fatty acids (PLFA) as lignin was subject of another recent review article
(Thevenot et al., 2010) and PLFAs are considered in such a large set of studies (c.f. Table 1)
that they would require a separate review. Finally GDGTs were excluded because their
application is predominantly in aquatic sediments rather than soils and they have been
recently reviewed (Schouten et al., 2013). This leaves the component classes labeled in bold
in Table 1 to be considered in the present study. Our study is relevant to the application of
compound-specific isotope analysis inasmuch that such analysis is directly affected by
variability and transformation of the underlying molecules. However, we did not explicitly
consider sources and effects of variation of the stable isotope signature of specific molecules
themselves, this being a research area of its own and also subject of recent review by
Diefendorf and Freimuth (2017). Furthermore, when considering application and preservation
of molecular proxies we restricted ourselves to topsoils (i.e. surface soil horizons = A
horizons as defined by the FAO in the Guidelines for soil description (2006)) as archives.
Paleosols as well as pedogenesis have been excluded as their formation and influence on the
preservation of molecular proxies forms an extensive research area in its own right that was
already the subject of another recent review article (Wiesenberg and Gocke, under review).
**2    Source related variability of molecular proxies**
**2.1    Definition**
Source related variability of molecular proxies pertains to intra-species variation in the
abundance of the molecules that are used as proxy. Such variability entails: i) variation in
relative abundance of individual compounds that together constitute the proxy, e.g. of *n*-
alkanes of different chain length in leaf waxes of a certain species; ii) variation in absolute
abundance of the molecules used as proxy either between different specimens or between
different parts of a single specimen. Depending on the research question, intra-species
variability of molecular proxies may be desirable or not. For instance when preserved leaf
wax lipids patterns are used to reconstruct past vegetation composition, the implicit



assumption is that the intra-species variability in the source vegetation is small compared to
the inter-species variability. In opposite, when the $\delta^2H$ signal of preserved leaf wax lipids is
used to reconstruct past precipitation patterns, one assumes that the precipitation induced
intra-species variability in the $\delta^2H$ patterns is large.
There are two main causes of intra-species variability in molecular proxies: internal variation
related to genetics and/or ontogeny; and external variation related to the growing
environment. Both are related in the sense that differences in response to environmental
factors are also often genetically determined (Shepherd and Griffiths, 2006). Here we discuss
both causes separately with a third paragraph devoted to studies were combined effects were
examined. For a detailed description of the biomolecular mechanisms of wax genesis and all
potential sources of change, the reader is referred to the review provided by Shepherd and
Griffiths (2006).
**2.2 Variation related to genetics and/or ontogeny**
**2.2.1 Wax lipids**
Many studies have indicated that the clear genetic control of leaf wax genesis leads to a
significant and meaningful difference in their composition (Shepherd et al., 1995; Shepherd
and Griffiths, 2006). For instance, prompted by the early works in this area (e.g. Eglinton et
al., 1962; Herbin and Robins, 1968; Herbin and Robins, 1969), Maffei performed an
extensive evaluation of the *n*-alkane patterns in several hundreds of plant species belonging
to the Gramineae, Umbelliferae, Cruciferae, Leguminosae, Cactaceae, Pinales, Lamiaceae,
Boraginaceae, Verbenaceae, Lolaneaceae and Scrophylariaceae (Maffei, 1994; Maffei,
1996a; Maffei, 1996b; Maffei et al., 1997; Maffei et al., 2004). These studies were
replenished by those on Styracacea (Li et al., 2013), Moraceae (Sonibare et al., 2005), and
Clusiaceae (Medina et al., 2004; Medina et al., 2006). Further, Dove et al. (1996) described
the alkane diversity among a grassland plant community, which enables tracing of the diet of
grazing animals due to the different alkane compositions of the plants. Recently, Mueller-
Niggemann and Schwark (2015) were able to differentiate rice from alternating crop plants
based on their *n*-alkane patterns. The results support the chemotaxonomic discriminatory
power of *n*-alkane patterns at family, sub-family and tribal level, which has been further
examined by Diefendorf et al. (2017). Examining plant *n*-alkane and *n*-alcohol distribution of
37 $C_4$ grasses, Rommerskirchen et al. (2006) also found chemotaxonomic differentiation was



possible at the sub-family level. Mongrand et al. (2001) examined the fatty acid composition
of the leaves of over 137 species of gymnosperms belonging to 14 families and collected
from different locations in France. They found a taxonomically meaningful clustering into
four main groups, with the highest discriminatory power in the Pinaceae at the genus level
(Mongrand et al., 2001). Additionally, Wiesenberg and Schwark (2006) determined
differences in the fatty acid composition between temperate $C_3$- and $C_4$-crops. Within the
same *Brassica* species of kale and swede Shepherd et al. (1995) observed a difference in
chain length distribution of wax lipids between two genotypes of the same species, indicative
of genetic control through variation in the enzyme system. Also for the isoprenoids, a
genetically driven discriminatory power related to (groups of) plant species is attributed
(Ohsaki et al., 1999; Jansen et al., 2007). However, an important issue is the phenotypic
plasticity of the genetic variability in leaf wax lipid patterns found and the implications
thereof for the stability of the patterns observed.
Maffei et al. (2004) concluded that phenotypic plasticity may overcome genetic variability,
particularly when plant developmental stages are considered along with abiotic and biotic
stress conditions. Several plant physiological studies have focussed on wax lipid composition
related to plant life stage, and report different results. Avato et al. (1984) found that where the
relative contribution of *n*-fatty acids, *n*-alcohols and *n*-alkanes differed between *Sorghum*
seedlings and mature leaves, the chain-length distribution within a component class remained
the same for the *n*-alkanes and *n*-alcohols. Giese (1975) observed a difference in homologue
dominance of *n*-alkanes between leaves of seedlings and mature barley plants. Also Herbin
and Robins (1969), Dyson and Herbin (1970), Baker and Hunt (1981), and Zhang et al.
(2004) identified increasing chain length dominance of leaf wax alkanes with increasing leaf
age. However, averaging of sampling over leaves of different age, position etc. within a stand
of trees did allow for distinction from other stands, indicating that inter-species variation was
larger than intra-species variation (Dyson and Herbin, 1970). Baker and Hunt (1981)
observed differences between adaxial and abaxial parts of leaves for some of the plant
species. Also Tulloch (1973) observed a variation of leaf waxes of several *Triticum* species
with age. In particular the whole plant *n*-alkane predominance shifted from $C_{31}$ at 24 days
after germination to $C_{29}$ at 100 days after germination (Tulloch, 1973). Furthermore,
Wiesenberg et al. (2004; 2012) and Wiesenberg and Schwark (2006) observed changes in *n*-
alkane and *n*-fatty acid compositions of a variety of temperate crop species with plant age.
Other publications reported seasonal variations in the *n*-alkane composition for variety of



pasture and crop plants by Dove et al. (1996), Hellgren and Sandelius (2001), Moseley
(1983), Shelvey and Koziol (1986) and various trees especially by Gülz and collaborators
(Prasad and Gülz, 1990; Gülz et al., 1991; Gülz and Muller, 1992; Gülz and Boor, 1992).
Variations in the alkane composition could be observed during the growing season among all
investigated plants, but general trends of increasing or decreasing chain length and *n*-alkane
contents have not consistently been determined. The *n*-alcohol predominance also varied but
to a much smaller extent, not affecting the predominance of a specific *n*-alcohol (Tulloch,
1973). Esters gradually showed an increase in esters of trans 2,3-unsaturated $C_{23}$ and $C_{24}$
acids with plant age (Tulloch, 1973). The variation was related to the development of the
plant, in particular that of flag leafs and sheets between 55 and 66 days (Tulloch, 1973).
Seldomly, also different source locations were analysed for their lipid composition, where the
plants could have developed specific lipid patterns. Kreyling et al. (2012) described
differences in the *n*-fatty acid and *n*-alkane composition of the same plant species originating
from different regions across Europe with different climatic conditions most likely due to
biosynthetic adaptation to the specific conditions.
In contrast to the previous, Li et al. (1997) studied the influence of ontogeny on leaf wax
lipids (*n*-alkanes, *n*-aldehydes, *n*-alcohols, esters, β-diketones, flavonoids and triterpenoids)
in several *Eucalyptus* species of the subgenus *Symphyomyrtus* on Tasmania, and found no
significant effect of ontogeny on leaf wax composition, which they found to clearly and
consistently differ between species (Li et al., 1997). Also Eglinton et al. (1962) observed that
the *n*-alkane composition of leaf waxes of 74 species of *Crassulacea* from the Canary Islands
showed no appreciable variation with respect to leaf position, age, size or specimen. Further,
Bush and McInery (2013) found no influence of canopy position or sampling time on the *n*-
alkane patterns of mature leaves from 24 tree species.

### 25    2.2.2   Cutin and suberin monomers

Cutin forms the molecular frame of the plant cuticle, whereas suberin is a cell wall
component of cork cells (Kolattukudy, 1981; Kögel-Knabner, 2002). As a result cutin occurs
mainly in the leaves of plants whereas suberin occurs on the outside of stems and roots of
woody plants, as well as in the endodermis and bundle sheet cells of grasses (Kögel-Knabner,
2002). Cutin and suberin monomers are mainly used as proxies to distinguish leaf from root
input in soils (Schreiber et al., 1999; Bull et al., 2000; Mendez-Millan et al., 2011) or as
proxy for related phenomena such as the degree of bioturbation in the topsoil (Nierop and



Verstraten, 2004). Therefore, the possible (onto)genetic effects on cutin and suberin
composition are a concern if they were to alter the composition of the polyesters to such an
extent that the separation between cutin and suberin is compromised.
Some general observations in literature are that long-chain even numbered $C_{20}$-$C_{30}$ $\omega$-hydroxy
fatty acids and $\alpha,\omega$-alkanedioic acids mainly originate from suberin, whereas shorter chained
$C_{16}$ and $C_{18}$ $\omega$-hydroxy fatty acids mainly derive from cutin (Schreiber et al., 1999; Otto et
al., 2005; Mendez-Millan et al., 2011). However, several publications challenge the universal
applicability of such general observations, indicating instead that genetic variability results in
many exceptions to such general rules. For instance, Hamer et al. (2012) found that $\omega C_{22:0}$,
$\omega C_{24:0}$ and $\omega C_{26:0}$ hydroxy fatty acids were not exclusively associated to roots, but also
occurred in the shoots of several species. In addition, $\omega C_{16:0}$ and $\omega C_{18:0}$ fatty acids were not
exclusive to the leaves, but also occurred in the roots of several species.

## 2.3   Variation related to environmental factors

### 2.3.1   Effects of temperature

Increased solar radiation levels are generally reported to lead to higher absolute amounts of
waxes produced (Sanchez et al., 2001; Shepherd and Griffiths, 2006). In addition, the
composition of the various component classes of wax lipids, i.e. the relative contribution of
$n$-fatty acids, $n$-alkanes, $n$-alcohols etc., has been reported to change. A shift towards lower
chain lengths within different component classes was sometimes found (Shepherd and
Griffiths, 2006). Thus, a positive correlation of long-chain odd $n$-alkanes with temperature
was observed (Maffei et al., 1993; Zhang et al., 2004). Also, the abundance of membrane
fatty acids with 16 and 18 carbons can change as a result of temperature (Maffei et al., 1993;
Williams et al., 1995; Matteucci et al., 2011). Often, under heat stress the relative abundance
of $C_{16:0}$ fatty acid was found to increase and vice versa the abundance of polyunsaturated
$C_{18:3}$ fatty acid to decrease (Larkindale and Huang, 2004; Bakht et al., 2006). Furthermore,
effects of temperature were observed for mono- and sesquiterpenes, with compounds like
limonene and myrcene having a close correlation with temperature, whereas others like 1,8-
cineol were not affected by temperature (Maffei et al., 1993). As a cause, a different
sensitivity of individual steps in the genesis of the wax lipid components is assumed
(Shepherd and Griffiths, 2006). However, results were found to vary between different
species and genotypes, indicating a species or genotype related sensitivity to changes in
irradiation (Shepherd and Griffiths, 2006), whereas cold- or heat-acclimated plants respond



differently than those that are not acclimated (Larkindale and Huang, 2004). Thus, a
dependency of temperature and lipid metabolism is widely observed, but especially in plants
other factors such as humidity or greenhouse gas composition might coincide with a larger
effect on the overall lipid composition.

### 2.3.2  Effects of humidity

With respect to the effects of water stress and/or high humidity, in their review Shepherd and
Griffith (2006) reported mixed results, with respect to absolute amounts as well as chain
length distribution. Bondada et al. (1996) reported an increase in absolute amounts of
epicuticular wax production by 69% in the leaves of cotton (*Gossypium Hirsutum* L.) under
water stress, which was confirmed by Hamrouni et al. (2001), Koch et al. (2006), Kim et al.
(2007), and Bettaieb et al.  (2010) for neutral lipids of other plant species. However, Kim et
al. (2007) found that water stress had only a minor effects on chain length distribution. The
relative contribution of different component classes to the wax composition remained
unchanged except for *Brassica oleracea* at the highest relative humidity, which showed an
increased contribution of ketones and primary alcohols and a reduction of secondary alcohols
and aldehydes (Koch et al., 2006). Recently, Srivastava et al. (2017) determined that
sustainable effects of drought on plant lipid composition are commonly missing with few
exceptions for perennial plants. Thus, several months after exposure to drought the lipid
biosynthesis and composition of leaves is resilient. The existing data shows that general
effects of drought on plant lipid composition are difficult to draw.

### 2.3.3  Effects of increased $CO_2$

Changes in greenhouse gases such as $CO_2$ have also been discussed to influence the lipid
biosynthesis and thus the lipid composition of plants. Short-term exposure of several hours to
elevated $CO_2$ concentrations e.g. during $^{13}CO_2$ or $^{14}CO_2$ labelling experiments has no or little
effect on the lipid composition, especially if sampling occurs several days after labelling
(Wiesenberg et al., 2009). In contrast a long-term rise in atmospheric $CO_2$ concentration has
been investigated in laboratory or free air carbon dioxide enrichment (FACE) experiments
(Ainsworth and Long, 2005). Although numerous such experiments have been maintained in
the meantime, implication of investigations of lipid composition is limited. Greenhouse
experiments showed that elevated $CO_2$ concentration affects the relative composition of
saturated and unsaturated fatty acids in wheat plants (Williams et al., 1994; Williams et al.,
1995; Williams et al., 1998). However, rising nitrogen fertilization and rising temperature can



lead to competing trends so that with elevated temperature and nitrogen fertilization
(Williams et al., 1995; Griepentrog et al., 2016). Although specific abundances of individual
long-chain alkanes and alcohols changed under elevated $CO_2$ concentration, the overall lipid
composition expressed as ACL and CPI did not change (Huang et al., 1999). Nevertheless,
concentration changes like an increase in *n*-alkane and *n*-alcohol abundances and a decrease
in *n*-fatty acid abundance was determined under rising $CO_2$ concentration, whereas nitrogen
fertilization led to a decrease in the effect (Huang et al., 1999), which was confirmed by
Wiesenberg et al. (2008a) for *n*-alkanes, *n*-fatty acids and *n*-alcohols. In some forest FACE
and open top chamber experiments, the effect of elevated $CO_2$ on plant lipid concentration
were not identified (Feng et al., 2010; Griepentrog et al., 2015), but the $^{13}CO_2$ labelling
associated with the $CO_2$ enrichment was used for tracing turnover of lipids in soils as
introduced by Wiesenberg et al. (2008b) for lipids.
## 2.4 Other or combined genetic, ontogenetic and/or environmental effects
Many studies considered the effects of e.g. geographical location on wax amounts and/or
composition without differentiating between individual genetic or environmental causes.
Again the exact parameters investigated vary greatly between studies, as do the conclusions
drawn. Cowlishaw et al. (1983) examined the *n*-alkane, *n*-alcohol, *n*-aldehydes and ester
composition of composite samples of four species of *Chionochloa*, one of which was sampled
at three different environmental locations to investigate environmental effects. They found
distinct chain length patterns that allowed for chemotaxonomic identification, where variation
between the three sampling sites did not alter dominant chain length patterns for any of the
component classes (Cowlishaw et al., 1983). Similar observations were made by Herbin and
Sharma (1969) for *ω*-hydroxy fatty acid composition of Pinus species from Asia, Europe,
North-America, Central America and the Caribbean. On the other hand, Piervittori et al.
(1996) found that the distribution of $C_{25}$, $C_{27}$, $C_{29}$ and $C_{31}$ *n*-alkanes in *Xanthoria parietina*
varied significantly between two different Piedmont valleys in Italy, and within those with
altitude, reflecting a combined influence of elevation, water availability, radiation and
temperature. For plaggen ecosystems Kirkels et al. (2013) also observed a significant
variability in reported ratios of the dominant *n*-alkanes with chain lengths $C_{27}$, $C_{29}$, $C_{31}$, $C_{33}$
most likely attributable to the causes examined here. However, in spite of this they found
meaningful clustering of the three different plant groups grasses, shrubs and trees indicating
that the variability did not obliterate the power of distinction (Kirkels et al., 2013). In a larger



study based on 2093 observations from 86 sources of plant material, Bush and McInerney
(2013) concluded that the general observation that $C_{27}$ and $C_{29}$ $n$-alkanes are dominant
markers for woody vegetation and $C_{31}$ for graminoids does not rigorously hold true. At the
same time $C_{23}$ and $C_{25}$ $n$-alkanes do seem to be robust indicators of *Sphagnum* (Bush and
McInerney, 2013) as already observed by Baas et al. (2000) and Pancost et al. (2002). Bush
and McInery (2013) indicated that the lack of rigour of the mentioned proxies is likely caused
by environmental conditions as indicated by a shift in patterns across African savannah and
rainforest environments.
The distinction between African savannah and rainforest environments in general and $C_3$
versus $C_4$ vegetation in particular have been the subject of more detailed research. Vogts et
al. (2009) studied the leaves and sometimes whole plants of 24 African rain forest and 45
savannah species. They found that as a result of environmental influence, including
temperature and aridity, chain length distributions of the $n$-alkanes and $n$-alcohols of some
species shifted to different chain length predominance. The environmental influences
overshadowed a taxonomic distinction at the order, family or sub-family level (Vogts et al.,
2009). Patterns in grasses were more consistent and thus less dependent on environmental
factors (Vogts et al., 2009). As a result, in spite of the environmental variability observed,
Vogts et al. (2009) found that by averaging lipid patterns within a given environment a clear
distinction between rain forest and savannah plants can be made, with a dominance of $C_{29}$ $n$-
alkane representative of the average rain forest plant signal and a dominance of $C_{31}$ $n$-alkane
of the savannah plants and $C_4$ savannah grasses. For the $n$-alcohols, $C_{28}$ dominated on
average for savannah plants, $C_{30}$ for rain forest plants and $C_{32}$ for $C_4$ savannah grasses (Vogts
et al., 2009).
Rommerskirchen et al. (2006) observed a generally higher content of $C_{31}$ and $C_{33}$ $n$-alkanes
and therefore higher ACL value in African $C_4$ grasses with respect to $C_3$ grasses from the
same area as a result of the genetic adaptation of $C_4$ grasses to warm, arid habitats. In
addition, $n$-fatty acid patterns have also been shown to vary with $C_3$ and $C_4$ metabolism, with
$C_3$ crops having relatively large proportions of $C_{24}$ $n$-fatty acid in leaves, stem and roots as
compared to $C_{22}$ and $C_{26}$ $n$-fatty acids in $C_4$ crops (Wiesenberg and Schwark, 2006).
**2.5   Conclusions and implications regarding source related variability**
Already Herbin and Robins (1969) concluded that there is a basic genetic control on the
composition of the wax components, including the alkanes, of plant leaves. However,





variable factors associated with age and environment can be superimposed upon the specific
pattern in some cases, while in others the genetically controlled pattern appears to be stable
and unaffected by external influences (Herbin and Robins, 1969). Now, 48 years later, a
much more extensive database has been accrued, albeit with a large emphasis on leaf wax
lipids in general and *n*-alkanes in particular. Nevertheless, the results are still equivocal. On
the one hand, there is ample evidence that genetically driven variability of leaf wax lipid
composition in principle leads to chemotaxonomically meaningful clustering that can form
the basis of the application of leaf wax lipids as molecular proxies. On the other hand, it is
clear that both ontogeny and environmental factors can have a significant and sometimes
dominant influence on lipid composition like e.g. chain length distribution. Matters are
complicated by the fact that much data with respect to the effects of environmental stress
originates from studies where plants were studied for a limited period of time (typically one
growing season), where extreme conditions were artificially imposed. In contrast, the lipid
signal from soil or sediment archives as used in reconstructions typically represents a mixture
of input of decades or longer from plants in various life stages of perennial plants, the
induced diversity of plants by frequent changes of annual plants in managed ecosystems and
the average of natural fluctuations in stress conditions during that time period.
In general from what is known to date, the conclusion seems justified that on the one hand
because of genetic and environmental influences there are no unique compounds nor 'golden
ratios' of different chain lengths of compounds that can always be linked to certain plants
under all circumstances. On the other hand, there are many situations where the influence of
genetic and environmental effects are small enough that they do not prevent the use of plant
lipids as molecular proxies. The currently available data does not allow for objective,
quantitative rules to be formulated in this respect. From the plant wax components, the *n*-
alkanes are the dominant class studied. In addition, research attention has focussed to a lesser
extent on *n*-alcohols and *n*-fatty acids. The other wax components such as isoprenoids and
ester bound lipids received hardly any research attention to date with respect to source related
variability in the context of their use as molecular proxies. Yet even for the *n*-alkane patterns
in leaf waxes, only a tiny portion of dominant plant species on the planet have been examined
in detail for the effects of genetics and environment on their amounts and patterns. It is clear
that much more research is needed in this respect.
Based on the current insights it seems prudent to explicitly take the possibility of genetically
and environmentally driven variability of lipid patterns into account when considering the use





of lipids as molecular proxies. For instance by considering plant species from the same
climatic zone as where the reconstruction takes place, and by mixing plant material from
different life stages to obtain the average molecular fingerprint to look for.

## 3  Input pathway related variability of molecular proxies

### 3.1  Definition

Here we discuss differences in the amount and composition of molecules used as proxies,
which is possible due to different input pathways of such molecules to the soil. A schematic
representation of the different input routes of molecular proxies into the soil is provided in
Fig. 1. The emphasis lies on potential effects for their use as molecular proxies. For a general
description of the different molecular origins of organic matter in soil, the reader is referred
to a dedicated review on this topic by Kögel-Knabner (2002).

### 3.2  Leaf versus root input

Conservative estimates calculate roots to represent 33% of global annual net primary
productivity (Jackson et al., 1997), whereas more recent studies highlight that the
contribution of root-derived organic matter in soils can account for >70% of total plant-
derived carbon (Rasse et al., 2005). As a result, roots form a considerable input of organic
matter in soils and are proposed to improve carbon storage in soils (Kell, 2012). In addition,
root input occurs to considerable depth in soils, ranging from an average depth of 0.5m in
tundra biomes to 15.0m in tropical grassland/savannah (Canadell et al., 1996). But also in the
temperate zone under certain circumstances such as the presence of nutrient rich fossil A
horizons at depth, deep rooting can be very significant (Gocke et al., 2015). However, on
average the majority of root biomass appears to be incorporated in the top 30 cm of the soil in
most biomes, i.e. in the topsoil (Jackson et al., 1996). The ratio of root/shoot biomass input is
also very variable across biomes, ranging from an average of 0.10 in cropland to 4.5 in
deserts (Jackson et al., 1996). Table 2 represents an overview of the average maximum
rooting depth, root biomass input in the first 30 cm of the soil and root/shoot biomass input
for different biomes (see also Fig. 1).
Therefore, if the molecules to be used as proxy are present in both leaves and roots of plants,
the possibility of root input is a factor that has to be considered depending also on the



purpose of the proxy. In the case of cutin and suberin monomers root input does not cause
interference as discerning root from leaf input is the specific purpose of this molecular proxy
(Mendez-Millan et al., 2011). However, this may be different for the wax lipids, i.e. *n*-
alkanes, *n*-alcohols, *n*-fatty acids and isoprenoids, that have been found to occur in leaves as
well as roots of species at varying concentrations (Jansen et al., 2007; Huang et al., 2011).
Particularly when such wax derived lipids are applied as molecular proxies for vegetation
cover in soil, root input can be an issue for two reasons: i) roots may contain a different wax
lipid composition than leaves qualitatively and quantitatively, thereby clouding the leaf signal
(Jansen et al., 2006; Martelanc et al., 2007); ii) young root input at depth may disrupt the
chronology of a reconstruction in time by overprinting the originally present signal (Lavrieux
et al., 2012; Gocke et al., 2014).
The main discussion with respect to the influence of root input in wax lipid based
environmental reconstructions from soils therefore revolves around assessing the relative
importance of root versus aboveground biomass input. Since plant wax lipids reside on the
outer parts of leaves and roots, relative surface area and bioproductivity are important. On a
global scale root surface area is almost always calculated to be higher than leaf surface area,
more than an order of magnitude so in grasslands (Jackson et al., 1997). However, in many
cases the absolute amount of lipids present per mass unit of root material is an order of
magnitude or more lower than on leaf material (Marseille et al., 1999; Zech et al., 2011). The
concurrent influence of such various factors makes the impact of root input a complex issue
that still is subject of scientific debate (Wiesenberg and Gocke, 2013).
Given that different factors will have a highly variable influence in different situations, no
general conclusion can be drawn. In some situations, the influence of roots as input pathway
of extractable lipids to be used as molecular proxy may be limited (Quenea et al., 2006). In
others, root input may be dominant (Van Mourik and Jansen, 2013). In addition, the relative
degree of influence may vary greatly with depth leading to the concurrent presence of leaf
lipid dominated and root lipid dominated zones at different depths in the same profile (Angst
et al., 2016).
**3.3   Microbial input**
In general, microbial biomass can be a significant source of soil organic matter, with up to
40% transformed to non-living soil organic matter, but is turned over much faster than plant
residues (Miltner et al., 2012). Focussing specifically on lipids, isotopic studies show that



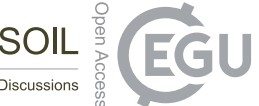

90% of fatty acids of microbial origin are turned over rapidly after cell death, whereas the
majority of biomass derived residual bulk C was stabilized in the non-living soil organic
matter pool (Kindler et al., 2009). In spite of the potentially shorter residence time, a
concurrent faster production makes that microorganism derived molecules are a factor to
consider when applying molecular proxies in soils except when such proxies are used to
study microbial input.
For wax lipids generally *n*-alkanes, *n*-alcohols and *n*-fatty acids with longer chain lengths
($>C_{20}$) and a distinct odd-over-even (*n*-alkanes) or even-over-odd (*n*-alcohols and *n*-fatty
acids) chain length predominance are considered to be higher plant derived, whereas shorter
chain length homologues are considered to be predominantly of microbial origin (Eglinton et
al., 1962; Dinel et al., 1990). Moreover, with the exception of an abundance of $C_{16}$ and $C_{18}$ *n*-
alcohol and *n*-fatty acid, such microbial lipids are described to lack a specific chain length
predominance (Stevenson, 1994; Lichtfouse et al., 1995). However, several researchers
challenge the observation that higher chain length lipids in soils are exclusively of higher
plant origin. Microorganisms have been shown capable of synthesizing higher chain length
straight-chain lipids, albeit usually to a limited extent (Ladygina et al., 2006; Nguyen Tu et
al., 2011). Jambu et al. (1978) indicated that while chain lengths $>C_{20}$ in soils are
predominantly plant derived, particularly in acidic soils fungi may contribute such lipids as
well. Furthermore, Marseille et al. (1999) observed an abundance of $C_{25}$ and $C_{27}$ *n*-alkanes
that they also attribute to *in-situ* production by fungi. This was confirmed for an agricultural
soil by Quenea et al. (2006), who observed old forest and fungi derived odd long-chain
alkanes based on compound-specific isotope analysis and lipid distribution patterns. Possible
pathways of *in-situ* genesis of *n*-alkanes in soils are reduction of *n*-alkenes and *n*-alcohols,
decarboxylation of bacterial *n*-fatty acids as well as degradation of biopolymers containing
aliphatic side chains (Lichtfouse et al., 1998). Nevertheless, based on the large number of
studies where typical higher plant derived patterns of lipids are reported and used in soils
(Table 1), including indicative ACL and CPI values, microbial input of longer chain length
straight-chain lipids generally does not seem to be a major factor compared to direct plant
derived input in the topsoil (Jansen and Nierop, 2009; Bai et al., 2009). In contrast, for
steroids and triterpenoids such as camposterol, stigmasterol and lupeol, microbial input in
soils can be considerable (Naafs et al., 2004). As another example, arbuscular mycorrhizal
fungi derived β-sitosterol is by far the most abundant sterol identified in soils (Grandmougin-
Ferjani et al., 1999).



With respect to cutin and suberin monomers, *in-situ* genesis in soils through microbial
transformation of other precursor molecules can be an issue. For instance, oxidation of free
fatty acids could be a source of $\omega$-hydroxy fatty acids, whereas microbial β-oxidation of
unsaturated fatty acids and/or mid-chain hydroxy fatty acids may be a source of $\alpha,\omega$-
alkanedioic acids, thus clouding the cutin/suberin signal (Naafs et al., 2004)
### 3.4   Airborne input
In addition to *in-situ* production and incorporation of soil lipids, airborne input must be
considered. The distance of airborne transport of larger constituents such as leaves can be
expected to be limited. However, smaller physical forms containing lipids, such as aerosols
and dust particles, can travel substantial distances (Conte and Weber, 2002) thus causing
input of alien molecules that may influence the local signal. This is of special importance
where airborne sediments with low content of organic matter are investigated as in these
environments already low inputs of foreign organic matter can significantly influence the
molecular proxies. Liu et al. (2007) showed that the $\delta^{13}C$ signature of sediment organic
carbon in loess deposits of the western Chinese Loess Plateau corresponds to that of dust
sources instead of the local vegetation. While in a study of marine sediment cores along the
Southwest African continental margin, Rommerskirchen et al. (2003) revealed that aerosol
derived input of higher chain-length *n*-alkanes and *n*-alcohols provides a significant signal,
the $\delta^{13}C$ signal of which corresponded well with continental C3/C4 plant distribution and
fossil pollen input when prevailing wind patterns were taken into account. However, in this
case, in contrast to vegetated soils, there was no *in-situ* input from higher plant vegetation.
Aerosol studies above plant canopies revealed a certain relationship of the plant wax
composition of the present plants, but significant differences from the biomass were observed
for *n*-alkanols and *n*-alkanes (Conte et al., 2003). While the wax molecular composition was
not directly linked between biomass and aerosol, especially the compound-specific isotope
composition ($\delta^{13}C$) revealed a closer link of both. For Bermuda aerosols it could be shown
that the aerosol compound-specific isotope composition of *n*-alcohols and *n*-acids reflects the
plant wax compound-specific isotope composition as well as the course of the bioproductivity
during the different seasons of the years (Conte and Weber, 2002).
In a study of $PM_{10}$ aerosols collected during a winter season in Baoij, China, Xie et al. (2009)
found concentrations of $C_{21}$-$C_{33}$ *n*-alkanes in the 10-600 $ng/m^3$ range as a result of intensive
coal burning in the region. In a two year study of $PM_{10}$ and $PM_{2.5}$ aerosols in urban sites in





Nanjing, Wang et al. (2006) observed $C_{21}$-$C_{33}$ *n*-alkanes present in the 10-100 ng/m$^3$ range.
Concentrations of $C_{21}$-$C_{35}$ *n*-alkanes in $PM_{10}$ aerosols in urban sites in Beijing sampled in all
seasons were even lower (Zhou et al., 2009). In this study also *n*-fatty acids and hopanes were
considered, but were found in small concentrations that, together with the *n*-alkanes,
constituted ca. 3% of the total organic matter in the aerosols (Zhou et al., 2009). In all studies,
the straight chain lipid patterns lacked the odd-over-even chain length predominance typical
of higher plants (Wang et al., 2006; Xie et al., 2009; Zhou et al., 2009). Nevertheless, in a
large survey a clear odd-over-even chain length predominance was found in spite of such
potentially intense aerosol derived input (Rao et al., 2011). This indicates that even in areas
under large aerosol deposition, as in the case of intensive anthropogenic pollution associated
with fossil fuel burning, the effect of aerosol deposition on *n*-alkane patterns in the soil is
limited as a result of the large *in-situ* input via roots and leaves of the local vegetation.

### 3.5    Conclusions and implications regarding input pathway related variability

The diversity of input pathways offers both opportunities and limitations for the use of
molecular proxies. Opportunities arise when different sources can be elucidated using
molecular proxies. Examples are the differences in molecular composition of leaf and root
waxes as used to differentiate between their respective influences, or when aerosol associated
lipids are used for source apportionment of terrestrial plant input in terrestrial or marine
sediments. This can help budgeting organic matter input of different sources and thus
improve (paleo-)environmental interpretations and reconstructions. Limitations are posed
when input through multiple pathways clouds the linkage of a (set of) molecule(s) to a certain
source for which it is to serve as proxy. For instance when linking a suite of straight-chain
lipids to a particular group of plants at a certain site. When looking at the application of
molecular proxies in soils, in particular the assessment of the influence of root derived input
is a challenge that is not always acknowledged. The significance of root derived organic
matter in soils and terrestrial sediments has been neglected for decades and has only been
recently highlighted (Rasse et al., 2005; Rumpel and Koegel-Knabner, 2011). More research
attention is needed to pinpoint how large possible interferences are and how the potential can
be to compensate for them, e.g. through modelling approaches. For instance, the VERHIB
model was designed to unravel the mixed *n*-alkane, *n*-alcohol and/or *n*-fatty acid signal
observed in soils into the most likely combination of plant groups responsible for the original



lipid input, treating leaves and roots explicitly as separate entities (Jansen et al., 2010). This
might form a starting point to disentangle leave and root derived lipid input.
Although the aerosol studies so far provide useful information that plant wax components are
transported via aerosols to remote places, other factors like degradation during transport and
integration of regional vegetation patterns may hamper direct source-to-sink relationship of
airborne molecular markers. Nevertheless the overall impact of aerosol borne molecules on
molecular proxy based reconstructions seems to be limited whenever the total abundance in
the soil is high.
**4   Transformations and turnover in soil**
Transformations and turnover of soil organic matter are an important study area in their own
right (Kögel-Knabner, 2002; Von Lützow et al., 2008). Important in the context of the
application of molecular proxies is the recent paradigm shift to the attribution of external
factors as drivers of soil organic matter turnover rates as opposed to inherent recalcitrance
related to molecular structure (Schmidt et al., 2011; Lehmann and Kleber, 2015). Coupled to
this are indications that microbial recycling of organic matter upon entering the soil
decouples the molecules from their biological sources (Miltner et al., 2012; Gleixner, 2013).
Here, we focus on the effects of (differences in) transformations/degradation of molecules in
soils for their use as molecular proxies. This includes transformations during the stages of
senescence or litter and covers attempts to estimate successive degradation processes of
organic matter occurring after burial until stages of long-term preservation (see also Fig. 1).
All of the attempts dealing with incorporation and preservation of organic matter deal with
different assumptions and entail different problems in terms of uncertainties. Thus, in
dependency of the environmental conditions, assumptions that are relevant for incorporation
and burial of organic matter play a major role, as should the different aspects of degradation
and preservation. However, currently much uncertainty exists regarding the influences of
individual environmental and genetic factors concerning degradation and preservation.
Therefore, the following paragraphs only provide the first insights tackling these issues,
which need further attention in future research projects.
Molecular transformations and variations thereof of molecular proxies mostly offer
complicate application of molecular proxies. However, in some instances they may also offer
opportunities. For instance, $n$-alkanes can be degraded to $n$-methyl ketones through β-



oxidation (Chaffee et al., 1986; Ambles et al., 1993), which can be used to assess and trace *n*-
alkane degradation in soils (Jansen and Nierop, 2009). Similarly, the presence of certain *seco*-
acids formed through A-ring opening of 3-oxytriterpenoids under anaerobic conditions, may
be used as proxy for the occurrence of such anaerobic episodes (Jaffe et al., 1996), e.g. under
stagnant water conditions.
## 4.1   Differences related to incorporation pathway
The incorporation pathway (Fig. 1) may influence subsequent turnover of molecular proxies.
This includes (differences in) degradation during senescence and/or litter degradation stages,
e.g. due to different input shapes (like root vs. leaf) offer a different degree of physical
protection.
In a study of *Gingko biloba* leaf wax lipids during the senescence and litter stages, Nguyen
Tu et al. (2003) found limited degradation that did not affect the dominant chain lengths of
alkyl molecular proxies. When comparing different classes of wax lipids they found the *n*-
alkanes to be the most resistant to degradation, followed by the *n*-fatty acids and then the *n*-
alcohols (Nguyen Tu et al., 2003). Also, more in general, in a study of grassland and forest
soils, Otto and Simpson (2005) determined that characteristic patterns of wax lipids and
isoprenoids were preserved throughout the stages between fresh plant material and soil
organic matter. They also determined preferential enrichment of suberin with respect to cutin
monomers in particular in one of the grassland soils (Simpson et al., 2008). This indicated for
example the fact that the former is embedded in woody tissue while the latter is exposed on
leaf surfaces (Simpson et al., 2008) (see also 4.3.3).
When looking at bulk organic matter in soils, Rasse et al. (2005) estimated that the main
residence time of root derived organic matter is on average 2.4 times that of shoot derived
organic matter. When comparing cutin and suberin monomers, Andreetta et al. (2013)
described selective preservation of leaf derived monomers in the more acidic and dryer soil,
while in the more fertile soil root derived monomers were preferentially preserved. They
attributed the former to inhibited microbial degradation due to drought and acidity, and the
latter to protection within aggregates. In another study still small differences in degradation
of the same *n*-alkanes that derived from different plants were found, with a slower
degradation of *n*-alkanes derived from more woody roots (Nierop and Jansen, 2009),
although lipids were generally well preserved. Killops and Frewin (1994) reported that





persistency of plant cuticles protected their composite isoprenoids from degradation in
mangrove sediments. Similar preservation in soils is also perceivable.
More in general, Mambelli et al. (2011) observed root litter, including biomarkers, to be
selectively preserved with respect to needle litter, which was confirmed by Mendez-Millan et
al. (2010) for maize and wheat roots versus shoots. Using isotopic signatures, Mendez-Millan
et al. (2011) were able to quantify and subsequently compensate for such differences in
turnover rate. This further emphasizes the significance of root derived organic matter for
turnover determinations as already discussed by Wiesenberg et al. (2004). In other words, the
relative abundance of roots and the uncertainties in terms of root related overprint in the
rhizosphere and rhizosphere extension entail large uncertainties and strong differences
between different plant species and environmental settings, especially at a molecular level.
Further research is required to enable extrapolations to or across ecosystem scales.
**4.2   Differences between different soil compartments**
When soils are used as archives of molecular proxies, mostly bulk samples are used and
replication per horizon or stratigraphic layer is often limited or absent. However, several
studies indicate that preservation of molecules used as proxies can differ between different
soil compartments (Flessa et al., 2008; Clemente et al., 2011; Griepentrog et al., 2014).
Depending on the research question this may pose a problem, for instance it might obscure
chronology when molecules are used as proxies to reconstruct changes over time.
Already Lichtfouse et al. (1998) showed that straight-chain lipids can become encapsulated in
larger humic polymers, thus being protected against degradation. In addition, physical
protection in (micropores of) aggregates and/or through association with clay minerals have
been identified as important pathways for stabilization of soil organic matter in general,
including molecules used as molecular proxies (Tonneijck et al., 2010). Using bulk and
compound-specific $\delta^{13}$C analysis, Cayet and Lichtfouse  (2001) showed that plant-derived $n$-
alkanes in a soil under maize cultivation varied in average age per particle size fraction, with
the $C_{31}$ $n$-alkane from the 200-2000μm fraction being significantly younger than that from the
50-200μm and 0-50μm fractions. A general trend of preferential preservation in smaller size
fractions, in particular the clay fraction, is also reported in other studies. For instance, Quenea
et al. (2004) and Flessa et al. (2008) observed longer turnover rates of soil organic matter in
smaller size fractions. Clemente et al. (2011) studied the preservation of long chain aliphatic
compounds in three soils with similar clay mineralogy but different carbon contents and



standing vegetation. Irrespective of these differences, they too found the aliphatic compounds to be preferentially preserved in the silt and clay fractions, and again linked this to strong interactions with the present clay minerals. In a recent study, Griepentrog et al. (2015, 2016) confirmed the higher residence time of organic matter in small sized density fractions when compared to macro-aggregates. This implies an improved preservation of organic matter associated with higher density and thus mineral association when compared to organic matter associated to lower density. However, physical fractionation techniques such as particle and density fractionation have a potential of creating analytical artifacts, especially when molecular proxies are investigated.

In addition, the effects of size or density fractions of soil on preservation of organic matter, including molecular proxies, are not uniform. For instance, Höfle et al. (2013) found size and density fraction related organic matter stabilization to be much less pronounced in the active upper layer than in the deeper soil horizons. This points to selective preservation of organic matter in the deeper soil because of more extensive aggregation and organo-mineral association. In a study of volcanic ash soils, Stewart et al. (2011) did not find differences in preservation of bulk soil organic matter in general or lipids in particular between different size fractions. They attributed this lack of differentiation to the presence of a large proportion of the soil organic matter that was not associated with mineral components as these were already saturated with previously incorporated soil organic matter (Stewart et al., 2011).

In general a combination of physical protection and sorptive preservation seems to be responsible for the observed differences (or lack thereof) in preservation of organic molecules in soils between different size or density fractions. This is corroborated amongst others by a study by Guggenberger et al. (1995), where they observed differences in the preservation of soil organic matter derived from tropical pastures compared to the preceding native savannah vegetation. They attributed this effect to a difference in interactions with the mineral phase, leading to physical protection of soil organic matter and molecular proxies contained therein. Similarly, differences in turnover rates between forest and grass derived molecules after land use change have been observed as a result of saturation of the adsorption sites on the mineral phase (Hamer et al., 2012).

In addition to heterogeneity in the effects of interactions with the mineral phase on preservation of molecular proxies, analytical artifacts cannot be completely excluded when physical and chemical fractionation techniques are applied to separate particle size or density fractions. To date systematic investigations addressing these issues are lacking, which

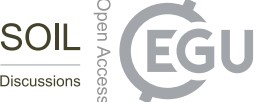

hampers the drawing of general conclusions with respect to processes that are relevant e.g.
under different climates and for different soil mineralogical composition.

### 4.3  Selective preservation within or between classes of molecules

Turnover rates of molecular proxies do not only vary between different compartments, but
may also vary within the same compartment; between and even within different (classes of)
molecules (Dinel et al., 1990; Bull et al., 2000; Amelung et al., 2008). For instance, Feng and
Simpson (2007) found preferential enrichment of straight-chain lipids as well as cutin and
suberin monomers with increasing depth with respect to bulk soil organic matter. In contrast,
in a study of grain-maize and silage-maize cropped soils Wiesenberg et al. (2004) found
turnover times in the sequence bulk soil organic matter > $n$-alkanes > $n$-fatty acids, with rate
differences that varied substantially between the two cultivations. The differences could be
related to the different biomass input on the one hand and large amount of lignite dust and the
low biomass input on the other hand, thus hampering degradation at this site. The faster
turnover of fatty acids than alkanes as also confirmed by Wiesenberg et al. (2008a) and
Griepentrog et al. (2015; 2016). In contrast, it may also offer opportunities to apply such
differences between molecular classes and their response to external factors to trace
transformations and input of organic matter in soils (Feng and Simpson, 2007).
An important issue with respect to the application of straight-chain lipids as molecular
proxies is also preferential degradation of certain chain lengths within a certain class of
molecules, as molecular ratios of various (higher) chain lengths are often used as proxies for
certain vegetation types (see paragraph 2). This issue is addressed in the following
paragraphs.

### 4.3.1  Straight-chain lipids

Already Moucawi et al. (1981) reported decreasing degradation rates with larger chain-length
for $n$-alkanes in soils, which was confirmed by Lichtfouse et al. (1998) who determined a
higher resistance of long straight-chain biopolymers in soil compared to their shorter chain
counterparts. However, such preferential degradation was found in agricultural and acidic
soils and in the absence of $Fe(OH)_3$ (Moucawi et al., 1981; Lichtfouse et al., 1998). Similar
results were found for other lipid classes as well (Moucawi et al., 1981). More recently,
several authors also indicate that such preferential degradation can occur in other soils
(Jansen and Nierop, 2009; Cui Jingwei et al., 2010). However, the extent of the effect





questions the suitability of the compounds in question as molecular proxies. For instance,
Jansen and Nierop (2009) found the overall effect of preferential degradation on higher plant
derived *n*-alkane patterns in soils to be small and not of influence for their use as vegetation
proxy. Similarly, Lei et al. (2010) determined that in spite of strong evidence of microbial
degradation, relative abundance of long-chain *n*-alkanes could still be used to distinguish
coniferous from broadleaf tree input in soils.
Within the group of straight-chain lipids, overall degradation rates of subclasses have been
found to vary depending on soil physicochemical properties. For instance, *n*-alkanes have
been reported to be better preserved in alkaline soils, whereas *n*-fatty acids accumulate in
more acidic soils (Simpson et al., 2008).
### 4.3.2 Isoprenoids
Isoprenoids are reported to have varying turnover rates both under oxic and anoxic conditions
in soils (Jaffe et al., 1996; Amelung et al., 2008). Generally, sterols, diterpenes and
pentacyclic triterpenes are reported to be turned over rapidly as compared to straight-chain
lipids in grassland as well as forest soils, hindering their application as molecular proxies for
their sources (Bull et al., 2000; Naafs et al., 2004; Jansen et al., 2007). However, Otto and
Simpson (2005) observed the exact opposite trend, indicating a strong environmental control
on the relative transformation rate of different classes of components. In an incubation study
of derived triterpenols, Koch et al. (2005) highlighted marked differences between
degradation rates of individual triterpenols, leading to a sharp relative increase in the
proportion of taraxerol with respect to the other triterpenols.
In addition, $\Delta^5$ sterols are transferred both aerobically and anaerobically to 5α- and 5β-stanols
(De Leeuw and Baas, 1986), which are reported to persist much longer in soils than their
precursors (Bull et al., 2000). Simpson et al. (2008) suggest to use the ratio of precursor
sterols to their stanol and stanone degradation products as measure for their degree of
degradation.
### 4.3.3 Cutin and suberin monomers
Bull et al (2000) observed different degradation rates for different components within the
classes of free and ester bound lipids, depending on soil chemical and physical composition.
However, Otto and Simpson (2006) found degradation of cutin and suberin to take place
without preference for specific constituents. In general, Quenea et al. (2004) described cutin





and suberin to be more resistant to degradation than free lipids residing in the same particle
size fraction.
In a study of hydrolysable lipids using compound-specific $^{13}$C analysis, Feng et al. (2010)
described mean turnover times for cutin and suberin derived ester-bound lipids of 32-34
years. While slower than for bulk soil organic matter in this system, it was much shorter than
anticipated, leading them to conclude that a large portion of cutin and suberin derived
compounds reside in the non-hydrolysable fraction (Feng et al., 2010).
As mentioned earlier (section 4.1), Simpson et al. (2008) observed preferential enrichment of
suberin monomers with respect to cutin monomers, which was confirmed by Mendez-Millan
et al. (2010). In addition to the physical location of suberin versus cutin as potential cause,
Simpson et al. (2008) suggested a higher resistance of suberin to degradation than cutin
owing to a larger content of phenolic units in the former. Mendez-Millan et al. (2010) argued
that microbial degradation, potentially influenced by the access to degradation sites are other
factors influencing the slower turnover of suberin vs. cutin monomers. Regardless of the
mechanism, the general difference in root vs. aboveground biomass derived suberin and
cutinin monomers and their individual turnover would clearly influence the application of the
cutin/suberin monomer ratio as proxy for leaf vs. root input.

## 4.4   Conclusions and implications regarding differences in transformations and turnover of molecular proxies in soils

Although available data is limited, it is clear that degradation of organic matter at a molecular
level in terrestrial archives such as soils, paleosols and sediments can significantly influence
the applicability of molecular proxies. As a result it seems useful to explore the possibility for
a correction to improve the determination of paleovegetation and vegetation shifts and other
paleoenvironmental information like paleotemperature and pH. The number of published
approaches to compensate for the influence of degradation on paleoenvironmental
reconstructions is still small. Zech et al. (2009) provided a simple two endmember model
approach to improve paleovegetation reconstruction based on molecular ratios of different
long-chain $n$-alkanes ($C_{27}$-$C_{33}$). Assuming that forest vegetation is dominated by $n$-$C_{27}$ alkane
and grass vegetation by $n$-$C_{31}$ and $n$-$C_{33}$ alkanes, high relative contributions of the respective
homologues of the assumed source vegetation are used as end-members. At the same time the
source vegetation is typically characterized by high odd-over-even predominance of long-
chain $n$-alkanes. On the other hand, soils reveal a low odd-over-even predominance and



abovementioned molecular ratios with smaller differences between the different vegetation
types. In theory, the degradation continuum from plant leaves to soils of the respective
vegetation type thus enable the identification of the degradation intensity of an unknown
sample, if the sample is mainly influenced by a single vegetation. If the unknown sample
does not plot on the degradation continuum, but between the different lines of different
vegetation types, the relative contribution of grass vs. tree derived vegetation might be
estimated and also corrected for the vegetation.
A slightly different approach was established by Buggle et al. (2010) who also used long-
chain *n*-alkane ratios and the odd-over-even predominance of alkanes for their correction.
While Zech et al. (2009) used correlations and then graphical-based reconstructions, Buggle
et al. (2010) used a calculation based approach. The degradation in the continuum from
recent soils is taken as an analogy and the slope of the regression line is multiplied with the
odd-over-even predominance and the addition of the intercept of a long-chain *n*-alkane ratio
in the crossplot of the ratio with the odd-over-even predominance. By moving the regression
line to an ancient sample set, the end of the regression line yields the former topsoil value of
the molecular ratio and odd-over-even predominance. Variation in the corrected long-chain *n*-
alkane ratio enable the assessment of fluctuations in palaeovegetation.
Both mentioned approaches rely on the general differentiation of grass vs. forest vegetation
based on long-chain *n*-alkane composition. As mentioned above such clear distinction of
vegetation types exclusively based on compounds deriving from one compound fraction such
as alkanes might be hampered by various factors such as variability within and between plant
species, thus leading to similar composition of e.g. alkane from coniferous trees and grass
plants (Maffei, 1996b; Maffei et al., 2004). Thus, such simple approaches might be
appropriate only in very well defined settings, where independent records such as pollen data
confirm the composition of specific plant assemblages determined by molecular proxies.
The expansion of approaches like the ones mentioned here to a broader range of molecular
proxies is required to receive more complete pictures and to acknowledge the different
turnover and degradation of different substance classes. However, the availability of datasets
on plant and soil chemical composition for substance classes other than the *n*-alkanes are
quite limited, hindering such expanding approaches. Thus, further surveys are required for
other molecular proxies than *n*-alkanes for a high diversity of plants and soils from different
climates. Afterwards, combined studies of more than one substance class enable improved
paleoenvironmental reconstructions, whereas cross-checking with other non-molecular





proxies, e.g. fossil pollen data, might be essential, especially if the paleorecord is targeted.
Also the extrapolation of such approaches to different environmental and climatic settings
might be limited as the effects of temperature, moisture, oxygen availability and others
influence the degradation of organic matter as discussed above. Consequently, proper
modelling approaches are required to assess not only palaeoenvironmental changes, but also
to acknowledge and identify degradation of organic matter at a molecular scale.

## 5  General conclusions

In this review we considered the three most important constraining factors for the application
of molecular proxies in soil science: i) variability in the molecular composition of plant
derived organic matter as a result of genetic or life stage variations or external environmental
factors; ii) variability in (relative contribution of) input pathways into the soil; and iii)
transformation and/or (selective) degradation of (some of) the molecules once present in the
soil. From the various studies done within and outside of soil science over the last decades
the following general picture emerges. All constraining factors considered can have a
significant influence on the applicability of molecular proxies in soil science. The degree of
influence of the constraining factors strongly depends on the type of molecular proxy as well
as the environmental context in which it is applied. In addition, the research question to be
addressed by application of the molecular proxy has a strong influence. A factor that poses a
constraining factor in one study might offer an opportunity in another. For instance microbial
degradation may constrain the application of molecular soil organic matter composition as
palaeo-vegetation proxy, but may offer the opportunity to study molecular transformation of
soil organic matter in the context of a study of soil carbon cycling. Recently, the first
modelling approaches to potentially compensate for some of the constraining factors,
specifically variability in input pathways and degradation of molecular proxies once in the
soil, have started to emerge. Based on the previous we strongly recommend that the potential
constraining factors are always explicitly considered whenever studies are planned in which
molecular proxies in soils play a role. This review may serve as starting point for gathering
the necessary information to decide, which constraining factors may play a role and how they
can be addressed best. At the same time, it became clear from available literature that much
information about the mentioned constraining factors is still lacking. In particular for
molecular classes other than *n*-alkanes, systematic information is often very scarce. We
therefore strongly appeal to the soil scientific community to address this knowledge gap. Also





for this our review may serve as a starting point with future applicability in soil science and
furthermore in paleopedology.

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



1    **Tables**

2    Table 1: Compounds frequently used as molecular proxies in soils

| Compound (the ones considerd in this review indicated in **bold**) | Most commonly used as proxy for: | Examples of recent publications[a]: | Number of articles published until 2017 *(publications 2007-2016)*[b] |
|---|---|---|---|
| Molecules of plant origin | | | |
| **n-alkanes, n-alcohols (n-alkanol), n-fatty acids (n-alkanoic acid)** | (groups of) plant species | (Zhang et al., 2006; Zeng et al., 2011; Jansen et al., 2013; Gocke et al., 2013) | alkane: 1588 *(1025)* <br> alcohol: 1972 *(1123)*; alkanol: 18 *(11)* <br> *n*-fatty acids: 43 *(27)*; *n*-alkanoic acid: 67 *(41)* |
| **n-methyl ketones** | degradation/transformation of soil organic matter | (Bai et al., 2006; Jansen and Nierop, 2009; Lei et al., 2010) | methyl ketone 104 *(50)* |
| **plant sterols and pentacyclic triterpenoids** | (groups of) plant species | (Volkman, 2005; Jansen et al., 2007; Lavrieux et al., 2011) | plant sterol: 1682 *(590)* <br> pentacyclic triterpenoid: 25 *(10)* |
| lignin monomers | coniferous species vs. broadleaf species vs. grasses and organic matter transformation | (Dignac et al., 2005; Nierop et al., 2006; Heim and Schmidt, 2007; Thevenot et al., 2010; | lignin monomer: 115 *(74)* |



| | | Simpson and Simpson, 2012) | |
|---|---|---|---|
| **cutin and suberin monomers** | root vs. aboveground biomass input | (Mendez-Millan et al., 2011; Hamer et al., 2012) | cutin monomer: 25 *(17)*<br><br>suberin monomer: 32 *(18)* |
| Molecules of animal or bacterial origin | | | |
| **Manure compounds such as coprostanol, 5β-stigmastanol, sitosterol and their epimers** | Human impact, animal husbandry | (D'Anjou et al., 2012; Birk et al., 2012) | coprostanol: 35 *(17)*<br><br>stigmastanol: 12 *(7)*<br><br>sitosterol: 70 *(47)* |
| glycerol dialkyl glycerol tetraethers (GDGT) | mean ambient air temperature, paleo-elevation and soil pH | (Luo et al., 2011; Weijers et al., 2011; Peterse et al., 2012; Ernst et al., 2013; De Jonge et al., 2014) | GDGT: 148 *(144)* |
| phospholipid fatty acids (PLFA) | microbial biomass | (Kramer and Gleixner, 2006; Kindler et al., 2009; Ngosong et al., 2012; Malik et al., 2013) | Phospholipid fatty acid: 2157 *(1628)*<br><br>PLFA: 1525 *(1140)* |
| Compound-specific stable isotope signal of one or more of the | | | |





| above[c] | | | |
|---|---|---|---|
| $\delta^{13}C$ | $C_3$ vs. $C_4$ plants and tracing carbon transformations e.g. by free air $CO_2$ enrichment (FACE) | (Sun et al., 2005; Feng et al., 2010; Mendez-Millan et al., 2012) | $^{13}C$: 13 *(11)* |
| $\delta^{15}N$ | (past) land management | (Bol et al., 2005; Griepentrog et al., 2014) | $^{15}N$: 2 *(2)* |
| $\delta^2H$ (deuterium) | precipitation and paleo-elevation | (Peterse et al., 2009; Bai et al., 2011; Luo et al., 2011; Sachse et al., 2012) | $^2H$: 6 *(4)* <br> deuterium: 9 *(7)* |
| $\Delta^{14}C$ (radiocarbon) | Age and contamination determination | Marschner et al., 2008; Mendez-Millan et al., 2014 | $^{14}C$: 3 *(1)* <br> radiocarbon: 35 *(30)* |

[a]Published from 2007 until 2017.
[b]According to ISI Web of Science, checked for 'soil' and 'target compound' in the topic of
articles on 27th February 2017 included in all available databases.
[c]'Compound-specific' and the respective isotope (i.e. $^{13}C$, $^{15}N$, $^2H$, and $^{14}C$ respectively) were
used as separate keywords in addition to 'soil'.



1    Table 2: average maximum rooting depth, biomass/depth distribution and root/shoot ratios in

2    different biomes (Canadell et al., 1996; Jackson et al., 1996)

| Biome: | Average maximum rooting depth: | Average percentage of roots in the top 30 cm: | Average root/shoot ratio: |
|---|---|---|---|
| Boreal forest | 2.0±0.3m | 83 | 0.32 |
| Cropland | 2.1±0.2m | 70 | 0.10 |
| Desert | 9.5±2.4m | 53 | 4.5 |
| Sclerophyllous shrubland and forest | 5.2±0.8m | 67 | 1.2 |
| Temperate coniferous forest | 3.9±0.4m | 52 | 0.18 |
| Temperate deciduous forest | 2.9±0.2m | 65 | 0.23 |
| Temperate grassland | 2.6±0.2m | 83 | 3.7 |
| Tropical deciduous forest | 3.7±0.5m | 70 | 0.34 |
| Tropical evergreen forest | 7.3±2.8m | 69 | 0.19 |
| Tropical grassland/savannah | 15.0±5.4m | 57 | 0.70 |
| Tundra | 0.5±0.1m | 93 | 6.6 |



**Figures**
Figure 1

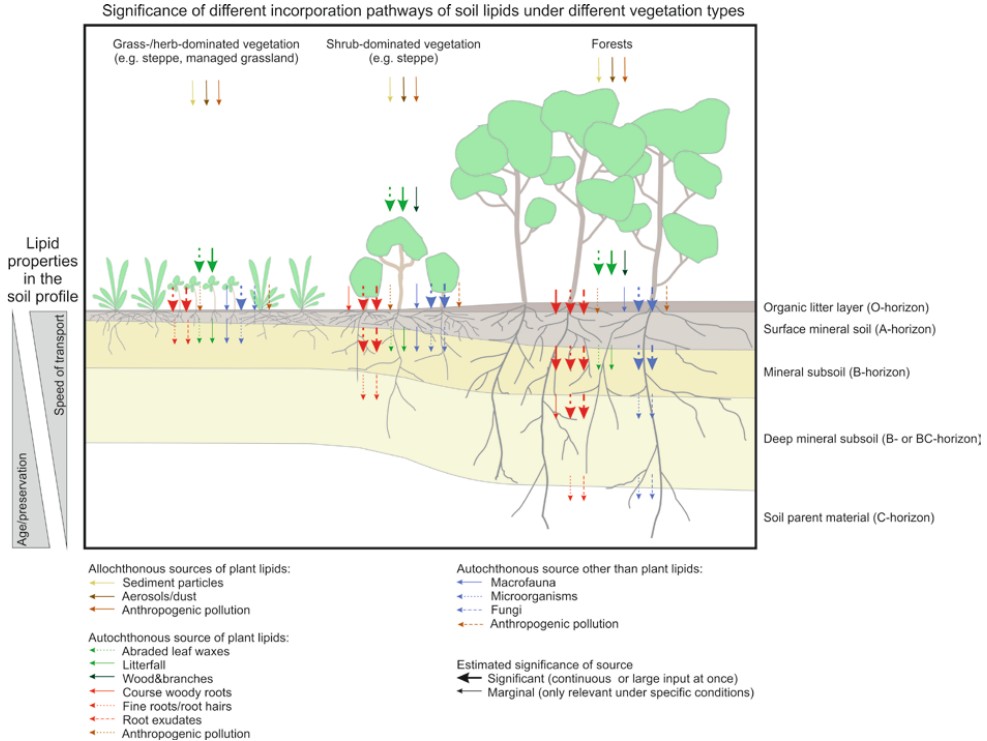

Figure caption
Conceptual overview of different incorporation pathways of lipids in soils originating from
different biological sources and anthropogenic contamination. The different sources are
indicated by distinct colors and lines of the arrows. The line thickness is an estimated
significance of individual sources, without providing quantitative measure for different
sources. Autochthonous sources are further distinguished by their significance in different
soil depths or soil horizons, respectively. Further, the transport and age/probability of
preservation as general properties of lipids are given at the left side of the figure.

