# Peer review of "Opportunities and limitations related to the application of"

_SOIL, 2017_

## Referee Comment (RC1) · Anonymous Referee #1 · 16 May 2017

The review article "Opportunities and limitations related to the application of plant-derived lipid molecular proxies in soil science" is overall an important and essential contribution to the SOIL community. However, I think some revisions are needed which would improve this review.

General comments:

-The introduction can be more compact. There are far too much direct quotes. You do not need to explain the word biomarker and molecular proxy in so much detail and how they are used for example in clinical studies. Keep the introduction short and simple, stay focused.

-In general, you should be more specific regarding the different biomarker groups and their strengths and weaknesses (e.g. the chemotaxonomic potential of the different leaf wax groups, see specific comments). Additionally, the chemotaxonomic potential of some plant-derived biomarker groups such as sterols or terpenoids is not discussed.

Specific comments:

Introduction

p3L15-23: Since you are doing your review with a specific focus on soils, this information is not necessary, you can delete it.

p3L24-30: I'm not sure if this information is really useful, but I think that's a matter of opinion.

p6L13-16: As far as I'm aware of, the review of Diefendorf and Freimuth (2017) is only about $\delta$13C, you may also refer to the review of Sachse et al. (2012) concerning $\delta$2H. Or you explicitly refer in line 14 to the stable carbon isotope signature.

p6L21: I cannot find this citation in the reference list. Can you provide at least the title and the journal where this article was submitted to? This would make it a lot easier for the interested reader to find the paper once it is published.

Section 2

p7L2-4: Since you have no intend to focus in your review on the $\delta$2H composition of plant biomarkers you may delete this sentence.

p7L9: Change the first were in where

p7-9 Section 2.2.1:

-You list some important studies that were done and that these studies prove a chemotaxonomic potential, but the section would benefit a lot if you would distinguish between studies that were done on fresh plant material and those done on soils (include some).

-For a soil scientist it would be interesting to know how the chemotaxonomic potential is transmitted to the soil and fortunately there exist some transect studies analysing this. Dig a bit in the literature, I'm quite sure you will find some.

-Maybe you can explain the results of the cited studies regarding the chemotaxonomic potential in more detail, i.e. which chain length represents which vegetation (at least in tendency, shorter chain length represents vegetation x, longer vegetation y). Even if you do not believe in the chemotaxonomic potential of the leaf waxes, you should state the difficulty in more detail that the reader can understand it.

p9L12-15: I think this belongs in the section where the environmental influences on the plant lipids are discussed.

p9L26-30: Here you describe the origin of the cutin and suberin monomers, but you have not stated the origin of the leaf waxes. I recommend to do it either for both or for none.

p10-15 Section 2.3:

-How are these changes transferred to the soil? How pronounced and over what period of time must these environmental changes occur that they can affect the overall leaf wax signal in the soil?

-Are there any environmental factors known to influence the cutin and suberin monomers?

p14L14-17: Good point! That's why from my point of view some studies regarding the chemotaxonomic potential of these compound classes in soils should be included in the section as well!

Section 3

p16L7: Is there a difference in the wax lipid distribution between roots and leaves? I know there is a quite interesting ongoing discussion whether one is able to distinguish

between root and leaf input using the patterns alone. Maybe you can address to this in more detail, e.g. by mentioning contradicting results of different studies (for example the study of Kirkels et al. (2013) observed general differences in the distribution between roots and leaves with a dominance of shorter chain lengths in roots compared to leaves while the study of Gocke et al. (2014) did not).

Also, is there a difference for different leaf wax groups (straight chain vs. cyclic compounds)? What about sterols and terpenoids?

p18L31: Is this a high input, does this contamination matter?

Section 4

p23L5-7: Why? Is there any explanation?

p23L27: What order are these differences?

p24L31-p25L6: Are there examples where degradation leads to a loss of the dominant compound?

p25L4: There are two Lei et al. 2010 in the reference list. Indicate if this is either B. Lei or G. Lei. Same for table 1.

Table 1

Column "Examples of recent publications"

You define recent as period from 2007-2017, but the publications mentioned in this column are not younger than 2014. Was there really nothing relevant published during the last three years? Either adapt recent or include at least one newer reference.

---

## Referee Comment (RC2) · Anonymous Referee #2 · 11 Jun 2017

A report for "Opportunities and limitations related to the application of plant-derived lipid molecular proxies in soil science"

Abstract

In general I think that the abstract is valid to be read in isolation although, as the subsequent review text, is excessively centered in alkyl compounds, whereas most main groups of cyclic biomarkers (terpenoids steroids,...) are neglected to large extent. In the same way, the importance of biodegradation and microbial synthesis in my opinion is overemphasized as regards: i) abiotic diagenetic transformations and ii) changes in the speciation status of extractable lipid molecules.

Line 15 and below: "Molecules used include extractable and ester-bound lipids as well as their carbon or hydrogen isotopic composition" – I would write more eclectically and prudently as regards the non extractable lipids. In fact, most of them are incorporated as esters, but the 'fixation' or 'immobilization' of lipids in the soil organo-mineral matrix or in the complex three-dimensional structure of soil organic matter include more or less efficient mechanisms (mainly in the case of compounds lacking reactive groups e.g., alkanes) such as hydrophobic bonding, diffusion intro microporous structures, solid solution, chemisorption not depending on esters but on reactive unsaturated double bonds, etc.

Line 19 and below (i, ii, iii) – I would perhaps extend this short list of "constraining factors" with additional points, or perhaps expanding point iii) "transformation and/or (selective) degradation of (some of) the molecules once present in the soil"... This would include the generic changes in solubility associated with the "speciation status" of formerly extractable lipid molecules. This in not included strictly into the term "transformation". Most lipids are subjected to a complex dynamic (to large extent abiotic) of polymerization and condensation together other organic and inorganic soil components. In most cases ,lipids turn into non-extractable compounds (polymerization of terpenes into macromolecular resins, photo-oxygenation and condensation or unsaturated aliphatic chains, etc) which are favored by soil desiccation and reactive colloidal minerals, and these nonextractable lipids may be stabilized in soil even in nonhydrolyzable forms.

In some cases it may be 'erroneously' considered that the lipids are biodegraded whereas the molecule remain intact in the soil a constituent of humic-type macromolecules, or encapsulated or entrapped into soil microcompartments where enzyme diffusion is largely hampered. On the other hand, changes in soil management or drastic environmental perturbation may lead to the release of lipid compounds (including pollutants) which were immobilized in the soil. The balance between the above processes may be responsible for a large proportion of the total variance in the composition of the molecular assemblages of lipid compounds in soil.

Apart from this, most of the research on (plant, soil) lipids has been based in GC/MS and a considerable amount of extractable lipid may consist on nonvolatile (high MW, oligomer) materials which could readily incorporate free lipids as a dynamic mechanism with a substantial bearing on the concentration and selective 'visibility' of lipid molecular proxies in soils (e.g., Soil Sci. 2001, 166(3), 186–196).

Other detail as regards the abstract and the whole review is that natural fires represent a frequent factor modifying the composition and the speciation status or lipid molecules, and some mention on these effects could be introduced in the review, which in my opinion is focused on biochemical processes, whereas abiotic reactions are also very active mainly in semiarid environments and soils under continental climate, where processes depending on the alternance of dry and wet seasons are crucial to explain lipid biogeochemical cycles.

Minor:

Page 1, line 16. I would rewrite "as well as" (I think that the subject of the sentence is "The molecules included" instead of e.g., "The information discussed. . .".

Page 2, line 15; page 12, line 16: It is often said that "parameter" is a misused word in English, that ought to been changed by "variable", "constraint", "factor", "index", measure", "characteristic". . .

Introduction

Page 2, line 25 and below: A frequent feature included in the definition of biomarker compounds is that they are produced exclusively by biosynthesis, and cannot be formed by abiotic reactions. This is often the most important feature in studies on extraterrestrial organic matter or in molecular paleontology studying Precambrian kerogens formed before the origin of Life.

Page 2, line 29 (and throughout the text) – Use "en"-dash for numeric ranges

Interactive
comment

[Figure]

Page 3, Line 26 and below: "...of the publications using molecular proxies in soil science have been published in the last ten years (2006-2015). On average ($\pm$ SEM) 59 $\pm$ 4 % of the publications with the respective keyword selections have been published in the last decade" – In fact, but depending on the searching strategy used by the authors, this could only be an affect of the fact that recent authors use 'new terms' to refer old subjects. This is also the case with, e.g., the so-called black carbon. There is a lot of classical (old) literature about this material, but the term was first used about 1982, and extensively used ca. 1993 (25 docs). Then, Scopus shows an increasing number of papers about this subject. In the same way, there is a lot of, in my opinion, interesting pioneer papers of soil lipid biomarkers which are not included in this review which, also in my opinion, include several recent papers not representing major contribution to classical studies . For instance, for this or similar paragraphs, I often cite the classical 1982 paper in Atmospheric Environment 16, 2139–159. There is also a specific paper of signature lipids in soil (not referred to as 'molecular proxies) in Eur. J. Soil Sci. 1996, 47, 183–196.

Page 4 (itemized list) : In general, I think that it is interesting to differentiate two main branches in the research lines on soil lipids: i) lipids as a source of biogeochemical information and, ii) lipids as active agents in soil processes reflected in soil quality; soil productivity, soil health...

Concerning the first line, there are several studies on the impact of forest fires on soil lipids. In fact, this is the environmental perturbation causing the major immediate and lasting effect on soil lipids; reviewed in e.g., (2004). Environment International (2004), 30: 855-870 or SJSS (2012), 2(2), 8-33 and references therein. In particular changes in diterpene resin acids and in RLC and CPI of alkanes are typical in fire-affected soils.

In general, and regarding the importance of biomarker compounds, or molecular tracers, or even lipid molecular fossils, there is large classical literature on the chemotaxonomic value of specific molecules in plants. I remember extensive use in the past of the classical series by e.g., Hegnauer, R. 1966. Chemotaxonomie der Pflanzen Bd.4.

Birkhauser Verlag. Basel und Stuttgart. Most important information in these old books (still?) cannot be retrieved readily using the 'modern' search in internet.

Concerning the effects of lipid on soils, there is extensive literature on the presumptive effect of lipid fractions in soil water repellency e.g., Geoderma (2010),155, 242-248 and Geoderma (2013) 206: 75-84, and references therein.

I would also include some paragraphs on the importance of several lipids compounds with an effect which may be allelopathic, antimicrobial, nematicidal, etc, and this have large importance in the soil organic matter transformation processes. There is many literature on the effect of resin acids (mono, di-sesquiterpenes. . .) from conifer plants as well as on the effect of several triterpenes in roots of angiosperms (amyrins, friedelans. . ..). All these compounds are also very important biomarkers, and in my opinion some paragraphs should highlight its importance.

Page 5, line 13: As suggested above, the polymerization and fixation of soil lipids would be an important natural phenomenon that cannot be considered included sensu stricto into the terms "transformation".

Page 6, lines 5–6. I consider that the mention to lignin is not necessary (it is not a major source of typical volatile lipid compounds).

Page 7.

I think that Botanists are changing too frequently names of classical plant taxa, and perhaps the most recent names ought to be used, for instance using Poaceae instead of Gramineae, Fabaceae instead of Leguminosae, etc.

Line 20: Brassicaceae instead of Cruciferae. Line 21: Scrophulariaceae instead of Scrophylariaceae. Line 21: Solaneaceae instead of Lolaneaceae (????). Line 32: Styracaceae instead of Styracacea.

Page 7. Note that in the list of taxons in lines 20–21 you combine plant families with plant orders (e.g., Pinales, the only family of this order being Pinaceae).

Page 9 line 21. Check: Crassulaceae instead of Crassulacea. Note that Latin names of families should not be italicized, as a difference with genus and species names.

Page 9, line 17: I think that most of the flavonoids are water-soluble compounds.

Page 9, Line 25: 2.2.2 Cutin and suberin monomers – Some authors prefer terms such as "structural units" or "units", in the case of macromolecules, and 'monomers' in the case of typical polymers.

Page 10, lines 7–12: Some mention could be done to the typical iso- and anteiso-branched fatty acids (and alkanes) mainly C15 and C17, for instance Microbiological Reviews (1991), 55, 288–302. Despite these branched chains have also been described in the uropygial gland of birds, they are frequently considered indicators of bacterial metabolism in soil.

Page 10, line 14 and below: I would improve (clairify) this important section on the effect of temperature: i) specifying better the paragraph about the effect of environmental temperature on lipid-synthesizing organisms and ii) the effect of T on the fate of lipids in the soil.

On the other hand, I would discuss more extensively the studies as regards effects of environmental temperature in the degree of saturation of the fatty acids, i.e., the fact that unsaturated FAs increase cell-membrane fluidity, and cell division, favoring the growing of organisms living in very cold environments.

Finally, and as indicated above, it would be interesting to explain some effects of extreme temperatures, i.e., the case with forest fires or controlled burning. I think that typical changes in diterpene resin acids and in alkyl series are were first reported in Geoderma, (1988). 42, 115 127.

Page 11, line 9: Do not capitalize the name of the species in "Gossypium hirsutum L.". I would not include the initial of Linnaeus due to it is not done in the case of the other species names in the Chapter.

Page 12, line 14: I would specify e.g., "cauliflower" in addition to "Brassica oleracea" (if this were the case). Depending on the variety, this species include very different plants (subspecies) with very different chemistry, e.g. broccoli, Brussels sprout, etc.

Page 12, line 23. Italicize Pinus. Page 12, line 18. I would add "the Poaceae", or "the graminaceous species", or "perennial grass" " before Chionochloa. Page 12, line 25. I would add "the lichen" before Xanthoria.

Page 15, line 20 and throughout the text: Leave blank space between numbers and units symbols

Page 17: I think this section is suitable to include some mention to brached fatty acids and alkanes, as the above indicated iso-and anteiso- chains. But also of chlorophyll-derived isoprenoids (phytane, pristane...) and other unusual in-chain branched alkenes typical of cyanobacteria.

Page 19: I think this section is suitable to include additional information on typical cyclic alkanes indicative of fossil organic matter or contamination with fossil fuels . This would be the case of hopanoids, but also of the classical chromatographic 'hump' of cyclic and branched alkanes generated by e.g., geothermal processes.

Page 20, subheading "Transformations and turnover in soil"

I think this would be a suitable section to be extended with additional subjects such as the "translocation of lipids" (amongst different soil microcompartments), a typical affect of forest fires, remember the classical studies in Proceedings. Soil Science Society of America (1970) 34:130—133.

It is also important the "speciation of lipids" in studies on factors involved in soil water repellence (Geoderma (2010) 155: 242–248) or in soils treated with urban wastes, biosolids, or sewage sludges (e.g., Waste Management & Research (2004) 22: 23–24) which systematically lead to a increase of extractable lipids in native soil organic matter, which is progressively incorporated and 'fixed' at different organizational levels

of the soil humic colloidal matrix.

Page 21, line 1: Amblès

Page 21, lines 2–4: Similarly, it is possible to take advantage of the seco- acids as markers of the impact of fire or thermal treatments in soils under pine vegetation, as suggested by classical literature e.g., The Journal of Organic Chemistry 1968., 33: 3718 3722, and Journal of the American Oil Chemists' Society, 1969. 46, 633 634.

In any case, I think that the pioneer booklet by Zinkel et al., (1971) ought to be indicated in the reviews (Diterpene Resin Acids. USDA, Forest Service. Forest Products Laboratory. Madison, Wisconsin. )

Page 22. Differences between different soil compartments

This section would be suitable to introduce some mention to 'lipid speciation' suggested in previous paragraphs. In particular the mechanisms leading to 'fixation' or 'immobilization" of lipids with the humic-type substances are extremely complex and effective, considering that compounds lacking reactive functional groups such as paraffins may be incorporated into condensed forms of organic matter, and require drastic chemical degradation methods for its release (e.g., Soil Biol. Biochem., (1987) 19(5): 513–520). Mechanisms not including encapsulation ("at a molecular level", line 20) but diffusion into microporous structures, or chemisorption in the case of unsaturated chains, fatty acids, etc, may be typical processes.

Note that not only "different soil compartments" but also "different plant compartments" are relevant as regards the distribution of lipids. The same alkane or fatty acid molecule may be present in extractive forms as an epicuticular plant wax. Or can be entrapped as nonhydrolyzable ester, or as sterically blocked alkyl chain in cutans, or in complex polyalkyl macromolecules of unknown structure presumptively existing in plants and soils, e.g., Naturwissenschaften (1986) 73: 579–585.

In the case of humic-type fractions, it has been indicated that condensation or polymerization of lipid may be an active abiotic mechanism (Naturwissenschaften (1991), 78: 359–362). Nevertheless, the pioneer studies were probably those from Australian soils (Aust. J. Soil Res. (1987) 25;71–82) suggesting a preferential incorporation of alkyl components in soil microaggregates (phenomenon which was also indicated in more recent papers cited in line 30).

Page 22, line 21. I prefer humic macromolecules or humic substances instead of 'polymers'. Page 23, line 1. I prefer alkyl rather than aliphatic (e.g., sugar is aliphatic).

Page 24, line 26. I would change 'biopolymers' (?) by alternative terms such as lipids, biomolecules, homologous series of-, etc (?).

Page 25, lines 1–11: As regards factors determining the presence or the lack of individual molecules in soils, please consider my above suggestions of not overemphasizing selective biodegradation of the lipid compounds, but also immobilization, condensation, insolubilization, fixation or encapsulation as molecules in 'recalcitrant', 'entrapped', 'fixed', or condensed forms, etc. . .

Page 28: General conclusions: I would discuss some of the above processes in the general conclusions

Line 20: I would change terms such as 'microbial degradation' by more general ones such as "microbial reworking". This will compensate the (relatively) ill-posed problem about the fate of lipids described in this chapter. Microorganims: i) degrade soil lipids, ii) synthesize alternative lipids which are released to soils, iii) contribute to the biodegradation of the organic matter with an affect of releasing its 'building blocks' and iv) modify the reactivity of organic matter by oxidative processes leading to e.g., increased titrable acidity (both carboxyl and phenolic hydroxyl groups), then humic-type organic matter have increased potential to retain, incorporates or insolubilize lipids into nonextractable forms.

References

Check the style of all references and abbreviations of Journals (e.g., page 30, line 20; page 31, line 9).

Italicize genus (Latin names), as in page 29, lines 23 and 29; page 32, line 12: page 40, line 8; page 42, line 7; page 45, line 2.

Page 30, line 2:italicize n- in n-alkanes. The same in page 31, line 5; page 32, line 28; page 33, line 9; page 39, line 19; page 43, line 11.

Page 31, line 6: Do not capitalize unnecessarily "isotopic". The same in some Title words in page 44, line 33.

Page 34. line 21. Subscript in CO2.

Page 41, line 34: Spectrom.

Page 44, line 5: Annu Rev Earth Planetary Sci.

---

## Author Comment (AC1) · 22 Jun 2017

Final author response:

We are grateful for the time and effort of the two referees. We are very glad to see the paper is deemed 'an important and essential contribution to the SOIL community' (referee 1). Upon the final verdict of the Topical Editor we will of course give a detailed response to all points raised by the two referees. For now in our final author response we shall give a more general response on the emerging issues and topics raised by the referees:

[Figure]

In response to Referee 1: - We agree there is potential for further condensation / sharpening of the focus of the paper. - Assessment of the chemotaxonomic potential of the molecular proxies as transferred to the soil is indeed very important. Therefore, we dedicated an entire section to it: section 4: "transformations and turnover in soils". We will more clearly link the results there to the chemotaxonomic potential at the source as discussed in section 2. Specific studies where the chemotaxonomic potential in soils is explored are already included (e.g. Jansen et al. palaeo-3, 2013), but will be highlighted more explicitly. Also the precise origin of the chemotaxonomic distinction (or lack thereof) will be more clearly explained. - The quantity and quality of extractable lipid patterns in leaf waxes vs. root waxes is indeed a topic of debate. We tried to capture this in section 3.2, but will make it even more explicit.

In response to Referee 2: - Our emphasis on straight-chain lipids is due to the fact that the vast majority of the work of molecular proxies, at least in the sense of chemotaxonomic application, has been on this compound class. We agree that this should be more clearly explained. - We do not agree that biodegradation and microbial synthesis are overemphasized. Both are important issues that should be considered when applying molecular proxies in soils. - The description of the ester-bound lipids is indeed somewhat imprecise. We will amend this in line with the suggestions of the referee. - The referee mentions various points with regards to transformation/degradation/preservation of lipids as part of soil organic matter (SOM) dynamics. We agree that the description of processes of transformation could benefit from further specification and we will critically re-read and amend/expand this section accordingly. However, it is explicitly not the aim or scope of the present article to enter a detailed discussion of molecular SOM dynamics. This is part of a separate on-going debate in the soil scientific community (e.g. Schmidt et al. Nature, 2011; Lehmann & Kleber Nature, 2015). For instance, the referee mentions molecules may remain intact as 'a constituent of humic-type macro-molecules'. The importance and even existence of such macro-molecules is currently under debate (Lehmann & Kleber, Nature, 2015). It is explicitly not our aim to contribute to this debate in our paper, as that issue alone

would be grounds for an entire review paper on its own. However, we will more explicitly mention the debate. - The referee asks for inclusion of more of the classical work on lipids. As can be seen in our reference list, we already went through great efforts to retrieve older and/or less accessible work. As a result we included a large body of older works including several publications in books (e.g. Eglinton et al., 1962; Herbin and Robins, 1968; Tulloch et al., 1973; Jambu et al., 1978; Tissot et al., 1984; Chaffee et al., 1986; Dinel et al., 1990).. However, strong focus of the review lies on more recent findings that generally helped to significantly change and improve our mechanistic understanding of processes influencing lipid composition in soil. Most of the relevant literature has been published during the last 2-3 decades, whereas older literature is often more descriptive on the one hand and on the other hand processes that were thought to be of high significance in the past and highlighted e.g. by Stevenson (1966, 1994) are now under debate, e.g. the concept of recalcitrance (Marschner et al., 2008; Dungait et al., 2012). Therefore, we focus more on the current state of knowledge.e Nevertheless, we will carefully evaluate the references suggested by the reviewer and include them where they are relevant. - The referee mentions a large number of other applications of molecular proxies in soils and the importance of other classes of components and additional forms of application than already mentioned in our review. For the sake of completeness we will carefully consider this. At the same time, focus is also needed (and specifically requested by referee 1!) to allow for in-depth discussion and to keep the review paper within acceptable page limits. Initially the review paper was substantially longer, but as per the request of the Editor was significantly reduced in length before it could be considered as discussion paper in SOILD. We cannot vastly expand the scope without concurrent vast reduction of depth and thoroughness. This is a fine balance that we wish not to disturb too much (but we will indicate our choices for delineation more clearly).

---

## Author Response (AR1)

**Rebuttal to referee comments "Opportunities and limitations related to the application of plant-derived lipid molecular proxies in soil science" – Boris Jansen & Guido Wiesenberg**

**Anonymous Referee #1**
The review article "Opportunities and limitations related to the application of plantderived lipid molecular proxies in soil science" is overall an important and essential contribution to the SOIL community. However, I think some revisions are needed which would improve this review.

General comments:
- The introduction can be more compact. There are far too much direct quotes. You do not need to explain the word biomarker and molecular proxy in so much detail and how they are used for example in clinical studies. Keep the introduction short and simple, stay focused.

*We condensed the introduction and removed some of the quotes. However, given the ambiguous and generic use of the term 'biomarker' and 'molecular proxy' we do feel that a thorough definition remains necessary.*

- In general, you should be more specific regarding the different biomarker groups and their strengths and weaknesses (e.g. the chemotaxonomic potential of the different leaf wax groups, see specific comments). Additionally, the chemotaxonomic potential of some plant-derived biomarker groups such as sterols or terpenoids is not discussed.

*We added the requested detail; see our response to the specific comments below.*

Specific comments:
Introduction
- p3L15-23: Since you are doing your review with a specific focus on soils, this information is not necessary, you can delete it.

*We do not agree entirely. As per the reviewers suggestion, we removed the links to medicine and toxicology (l. 4-6, p. 3) as being perhaps too broad. However, we do feel thatit is important to place the use of molecular proxies in soil science firmly within the broader context of the application of molecular proxies across different fields of science. This is relevant, as these fields of science can learn and profit from one another but for this they must be aware of each other's use of molecular proxies. For instance, as explained later in the introduction and extensively referenced in the body chapters of the manuscript, there is much plant physiological literature that includes information crucial for the application of molecular proxies in soil science. It is important that those using molecular proxies in soil science are aware of the broader application of such proxies in other fields.*

- p3L24-30: I'm not sure if this information is really useful, but I think that's a matter of opinion.

*We deleted the specific Scopus analysis.*

- p6L13-16: As far as I'm aware of, the review of Diefendorf and Freimuth (2017) is only about _13C, you may also refer to the review of Sachse et al. (2012) concerning _2H. Or you explicitly refer in line 14 to the stable carbon isotope signature.

*We added the reference to Sachse et al. and changed the wording.*

- p6L21: I cannot find this citation in the reference list. Can you provide at least the title and the journal where this article was submitted to? This would make it a lot easier for the interested reader to find the paper once it is published.

*We removed this reference as the article in question unfortunately is still not published.*

Section 2
- p7L2-4: Since you have no intend to focus in your review on the _2H composition of plant biomarkers you may delete this sentence.

*Deleted as requested*

- p7L9: Change the first were in where

*Changed as requested*

- p7-9 Section 2.2.1:
  -You list some important studies that were done and that these studies prove a chemotaxonomic potential, but the section would benefit a lot if you would distinguish between studies that were done on fresh plant material and those done on soils (include some).
  -For a soil scientist it would be interesting to know how the chemotaxonomic potential is transmitted to the soil and fortunately there exist some transect studies analyzing this. Dig a bit in the literature, I'm quite sure you will find some.
  -Maybe you can explain the results of the cited studies regarding the chemotaxonomic potential in more detail, i.e. which chain length represents which vegetation (at least in tendency, shorter chain length represents vegetation x, longer vegetation y). Even if you do not believe in the chemotaxonomic potential of the leaf waxes, you should state the difficulty in more detail that the reader can understand it.

*We are of course aware of the chemotaxonomic potential of plant waxes preserved in soils and have successfully exploited this potential in our own work as evidenced by several publications that were also cited later on in the present study (e.g. Jansen et al. 2010; Jansen et al. 2013). We expanded this section with the requested information and included reference to these and some other articles by others who successfully used molecular proxies for chemotaxonomic differentiation in soils.*

- p9L12-15: I think this belongs in the section where the environmental influences on the plant lipids are discussed.

*We agree. For the sake of condensation, and because the environmental influences are already extensively discussed and referenced in section 2.3, we removed these lines here.*

- p9L26-30: Here you describe the origin of the cutin and suberin monomers, but you have not stated the origin of the leaf waxes. I recommend to do it either for both or for none.

*We removed the description of the origin of cutin and suberin monomers here.*

- p10-15 Section 2.3:
  -How are these changes transferred to the soil? How pronounced and over what period of time must these environmental changes occur that they can affect the overall leaf wax signal in the soil?

-Are there any environmental factors known to influence the cutin and suberin monomers?

*Both are very good questions to which the answers remain equivocal. We explicitly address this in section 2.5 where we draw general conclusions about the influences (plasticity, environmental) that were discussed in section 2.*

- p14L14-17: Good point! That's why from my point of view some studies regarding the chemotaxonomic potential of these compound classes in soils should be included in the section as well!

*The requested information has now been included in this section (see our previous comments).*

Section 3
- p16L7: Is there a difference in the wax lipid distribution between roots and leaves? I know there is a quite interesting ongoing discussion whether one is able to distinguish between root and leaf input using the patterns alone. Maybe you can address to this in more detail, e.g. by mentioning contradicting results of different studies (for example the study of Kirkels et al. (2013) observed general differences in the distribution between roots and leaves with a dominance of shorter chain lengths in roots compared to leaves while the study of Gocke et al. (2014) did not). Also, is there a difference for different leaf wax groups (straight chain vs. cyclic compounds)? What about sterols and terpenoids?

*Several studies have looked at straight chain lipid composition in roots and leaves and found significant differences in quantity (generally lower in roots than leaves per gram of dried material) and composition. With respect to the latter, differences vary between species, but are generally observed to be so large that the difference between the leaves and roots of a certain species are of similar magnitude as the differences between the leaves of two separate species. We added the requested detail to the text here. The suggestion to use the difference between root and leaf patterns to potentially separate their input in soil archives was already explicitly mentioned in the concluding paragraph of this section (p19 l.29 – p20 l.2).*

*Much less is known about other leaf wax compound classes than straight-chain lipids to the point that we feel we cannot confidently discuss this here.*

- p18L31: Is this a high input, does this contamination matter?

*Generally, n-alkane concentrations in fresh plant material are in the µg/g dried material range, i.e. a factor 1000 higher. However, the signal preserved in the soil strongly depends on the input and preservation of the leaf derived organic matter. With typical organic C concentrations in the 1-5% ranges in many soils, contamination such as mentioned here could be a significant factor.*

Section 4
- p23L5-7: Why? Is there any explanation?

*The authors of the cited articles link it to interaction with the mineral phase. This is now included in the text.*

- p23L27: What order are these differences?

*The authors of the cited article (Hamer et al. 2012) found turnover times of the labile pool of ca. 4.4 years, whereas the stable pool turned over in the decadal scale leading to approximately one order of magnitude difference. This has been added to the text.*

- p24L31-p25L6: Are there examples where degradation leads to a loss of the dominant compound?

*To the best of our knowledge selective degradation such that the dominant compound is lost, whereas the other chain lengths are not affected, does not occur. However, the studies specifically focusing on this effect are few, so we cannot confidently claim this in the present review article.*

- p25L4: There are two Lei et al. 2010 in the reference list. Indicate if this is either B. Lei or G. Lei. Same for table 1.

*Corrected as requested.*

- Table 1 Column "Examples of recent publications" You define recent as period from 2007-2017, but the publications mentioned in this column are not younger than 2014. Was there really nothing relevant published during the last three years? Either adapt recent or include at least one newer reference.

*We added several more recent references and deleted some of the oldest in the table. We expanded the definition of 'recent' to include 2005-2017 because for some of the more rarely used proxies indeed the number of publications is low.*

**Anonymous Referee #2**
A report for "Opportunities and limitations related to the application of plant-derived
lipid molecular proxies in soil science"

Abstract
- In general I think that the abstract is valid to be read in isolation although, as the
  subsequent review text, is excessively centered in alkyl compounds, whereas most
  main groups of cyclic biomarkers (terpenoids steroids,. . .) are neglected to large
  extent. In the same way, the importance of biodegradation and microbial synthesis in
  my opinion is overemphasized as regards: i) abiotic diagenetic transformations and ii)
  changes in the speciation status of extractable lipid molecules.

*A clear and motivated delineation of the compound classes considered in our review is given
in the introduction (p. 4, l.29 – p. 6, l. 21). These explicitly include the cyclic biomarkers
mentioned. It is true that the alkyl compounds receive more attention in the review than
some of the other classes of components. This is borne out of necessity, as the majority of
research concerning biomarkers in soils focusses on alkyl compounds (see also our reply to
the last comment of Referee #1). We explicitly acknowledged the unequal distribution of
research efforts with respect to different compound classes on multiple occasions, and urged
for future research to help overcome this bias (e.g. p.14, l.24-26; p. 27, l.26-32). With respect
to the potential mechanisms of disturbance to consider, clearly a choice had to be made to
keep the manuscript within reasonable page limits. We therefore made a motivated choice to
focus on three main processes and underpinned this with multiple literature references (p. 5,
l. 1-16).*

- Line 15 and below: "Molecules used include extractable and ester-bound lipids as
  well as their carbon or hydrogen isotopic composition" – I would write more
  eclectically and prudently as regards the non extractable lipids. In fact, most of them
  are incorporated as esters, but the 'fixation' or 'immobilization' of lipids in the soil
  organo-mineral matrix or in the complex three-dimensional structure of soil organic
  matter include more or less efficient mechanisms (mainly in the case of compounds
  lacking reactive groups e.g.,alkanes) such as hydrophobic bonding, diffusion intro
  microporous structures, solid solution, chemisorption not depending on esters but on
  reactive unsaturated double bonds, etc.

*We changed the wording as requested to emphasize that non-extractable lipids include more
than ester-bound lipids alone.*

- Line 19 and below (i, ii, iii) – I would perhaps extend this short list of "constraining
  factors" with additional points, or perhaps expanding point iii) "transformation
  and/or(selective) degradation of (some of) the molecules once present in the soil". . .
  This would include the generic changes in solubility associated with the "speciation
  status" of formerly extractable lipid molecules. This in not included strictly into the
  term "transformation". Most lipids are subjected to a complex dynamic (to large
  extent abiotic) of polymerization and condensation together other organic and
  inorganic soil components. In most cases ,lipids turn into non-extractable compounds
  (polymerization of terpenes into macromolecular resins, photo-oxygenation and
  condensation or unsaturated aliphatic chains, etc) which are favored by soil
  desiccation and reactive colloidal minerals, and these nonextractable lipids may be
  stabilized in soil even in nonhydrolyzable forms

*In our view processes such as condensation and polymerization are captured under the term
'transformation'. We completely agree that processes as described by the referee are crucial
within the context of soil organic matter (SOM) dynamics. However, as stated on p. 20 l. 11-
29 at the onset of section 4, such processes would warrant a review paper of its own and*

*have indeed been subject of separate review in the past. In addition, they are part of a separate on-going debate in the soil scientific community (e.g. Schmidt et al. Nature, 2011; Lehmann & Kleber Nature, 2015). It is explicitly not the aim or scope of the present article to enter a detailed discussion of molecular SOM dynamics. Instead we limited ourselves to potential effects of transformations on the applicability of molecular proxies only.*

- In some cases it may be 'erroneously' considered that the lipids are biodegraded whereas the molecule remain intact in the soil a constituent of humic-type macromolecules,or encapsulated or entrapped into soil microcompartments where enzyme diffusion is largely hampered. On the other hand, changes in soil management or drastic environmental perturbation may lead to the release of lipid compounds (including pollutants) which were immobilized in the soil. The balance between the above processes may be responsible for a large proportion of the total variance in the composition of the molecular assemblages of lipid compounds in soil. Apart from this, most of the research on (plant, soil) lipids has been based in GC/MS and a considerable amount of extractable lipid may consist on nonvolatile (high MW, oligomer) materials which could readily incorporate free lipids as a dynamic mechanism with a substantial bearing on the concentration and selective 'visibility' of lipid molecular proxies in soils (e.g., Soil Sci. 2001, 166(3), 186–196).

*The very existence of humic-type macromolecules is currently under debate (Lehmann & Kleber Nature, 2015: see our reply to the previous comment). That said, we now expanded the text to explicitly include the possibility of extractable lipids transforming to non-extractable forms, regardless of the underlying process and emphasized the possibility of occlusion hampering extraction and detection. For this we also included the reference suggested by the referee.*

- Other detail as regards the abstract and the whole review is that natural fires represent a frequent factor modifying the composition and the speciation status or lipid molecules, and some mention on these effects could be introduced in the review, which in my opinion is focused on biochemical processes, whereas abiotic reactions are also very active mainly in semiarid environments and soils under continental climate, where processes depending on the alternance of dry and wet seasons are crucial to explain lipid biogeochemical cycles.

*While we needed to limit the scope of the article for the reasons previously outlined, we agree that the effects of natural fires should be included. Therefore, we now added a brief discussion of the effects of natural fires in the introduction of section 4 as well as in section 4.2 where we now specifically address the potential effects on both quantity and quality of molecular proxies.*

Minor:
- Page 1, line 16. I would rewrite "as well as" (I think that the subject of the sentence is "The molecules included" instead of e.g., "The information discussed. . .".

*The sentence was changed*

- Page 2, line 15; page 12, line 16: It is often said that "parameter" is a misused word in English, that ought to been changed by "variable", "constraint", "factor", "index", measure", "characteristic". . .

*Changed as requested*

- Introduction Page 2, line 25 and below: A frequent feature included in the definition of biomarker compounds is that they are produced exclusively by biosynthesis, and cannot be formed by abiotic reactions. This is often the most important feature in studies on extraterrestrial organic matter or in molecular paleontology studying Precambrian kerogens formed before the origin of Life.

*This section was condensed as per the suggestion of Referee #1.*

- Page 2, line 29 (and throughout the text) – Use "en"-dash for numeric ranges

*Changed as requested*

- Page 3, Line 26 and below: ". . .of the publications using molecular proxies in soil science have been published in the last ten years (2006-2015). On average (± SEM) 59 ± 4 % of the publications with the respective keyword selections have been published in the last decade" – In fact, but depending on the searching strategy used by the authors, this could only be an affect of the fact that recent authors use 'new terms' to refer old subjects. This is also the case with, e.g., the so-called black carbon. There is a lot of classical (old) literature about this material, but the term was first used about 1982,and extensively used ca. 1993 (25 docs). Then, Scopus shows an increasing number of papers about this subject. In the same way, there is a lot of, in my opinion, interesting pioneer papers of soil lipid biomarkers which are not included in this review which, also in my opinion, include several recent papers not representing major contribution to classical studies . For instance, for this or similar paragraphs, I often cite the classical 1982 paper in Atmospheric Environment 16, 2139–159. There is also a specific paper of signature lipids in soil (not referred to as 'molecular proxies) in Eur. J. Soil Sci. 1996, 47, 183–196.

*This section specifying publications in the last ten years was removed as per the suggestion of Referee #1. With respect to older literature, as can be seen in our reference list, we already went through great efforts to retrieve older and/or less accessible work. As a result we included a large body of older works including several publications in books (e.g. Eglinton et al., 1962; Herbin and Robins, 1968; Tulloch et al., 1973; Jambu et al., 1978; Tissot et al., 1984; Chaffee et al., 1986; Dinel et al., 1990). However, strong focus of the review lies on more recent findings that generally helped to significantly change and improve our mechanistic understanding of processes influencing lipid composition in soil. Most of the relevant literature has been published during the last 2-3 decades, whereas older literature is often more descriptive on the one hand and on the other hand processes that were thought to be of high significance in the past and highlighted e.g. by Stevenson (1966, 1994) are now under debate, e.g. the concept of recalcitrance (Marschner et al., 2008; Dungait et al., 2012). Therefore, we chose to include older work, but focus more on the current state of knowledge.*

- Page 4 (itemized list) : In general, I think that it is interesting to differentiate two main branches in the research lines on soil lipids: i) lipids as a source of biogeochemical information and, ii) lipids as active agents in soil processes reflected in soil quality; soilproductivity, soil health. . . Concerning the first line, there are several studies on the impact of forest fires on soil lipids. In fact, this is the environmental perturbation causing the major immediate and lasting effect on soil lipids; reviewed in e.g., (2004). Environment International (2004), 30: 855-870 or SJSS (2012), 2(2), 8-33 and references therein. In particular changes in diterpene resin acids and in RLC and CPI of alkanes are typical in fire-affected soils.

*A discussion of the effects of natural fires is now included (see our reply to the previous comments).*

- In general, and regarding the importance of biomarker compounds, or molecular tracers, or even lipid molecular fossils, there is large classical literature on the chemotaxonomic value of specific molecules in plants. I remember extensive use in the past of the classical series by e.g., Hegnauer, R. 1966. Chemotaxonomie der Pflanzen Bd.4. Birkhauser Verlag. Basel und Stuttgart. Most important information in these old books (still?) cannot be retrieved readily using the 'modern' search in internet.

*See our reply to the previous comments.*

- Concerning the effects of lipid on soils, there is extensive literature on the presumptive effect of lipid fractions in soil water repellency e.g., Geoderma (2010),155, 242-248 and Geoderma (2013) 206: 75-84, and references therein.I would also include some paragraphs on the importance of several lipids compounds with an effect which may be allelopathic, antimicrobial, nematicidal, etc, and this have large importance in the soil organic matter transformation processes. There is many literature on the effect of resin acids (mono, di-sesquiterpenes. . .) from conifer plants as well as on the effect of several triterpenes in roots of angiosperms (amyrins, friedelans. . ..). All these compounds are also very important biomarkers, and in my opinion some paragraphs should highlight its importance.

*Of course there is much more to be said about the diversity of lipids in soils and SOM transformation in soils than is included in the present review. However, SOM transformation itself explicitly is not the focus of this review and is only concisely addressed in the context of its effect on molecular proxies. Even then of course one can always discuss whether or not more information should be included. However, in the light of the fact that we already had to seriously condense the manuscript before it was accepted for publication as discussion paper on SOILD, and even now Referee #1 asked for further condensation, we really cannot significantly expand the discussion on SOM transformations.*

- Page 5, line 13: As suggested above, the polymerization and fixation of soil lipids would be an important natural phenomenon that cannot be considered included sensu stricto into the terms "transformation".

*See our previous response to the general comment by the referee on this topic.*

- Page 6, lines 5–6. I consider that the mention to lignin is not necessary (it is not a major source of typical volatile lipid compounds).

*We disagree. As indicated in our extensive definition of molecular proxies in the introduction, such proxies are not limited to typical volatile lipid compounds. In fact we explicitly include and discuss cutin and suberin in the review paper, neither of which is a volatile compound.*

- Page 7. I think that Botanists are changing too frequently names of classical plant taxa, and perhaps the most recent names ought to be used, for instance using Poaceae instead of Gramineae, Fabaceae instead of Leguminosae, etc. Line 20: Brassicaceae instead of Cruciferae. Line 21: Scrophulariaceae instead of Scrophylariaceae. Line 21: Solaneaceae instead of Lolaneaceae (????). Line 32: Styracaceae instead of Styracacea.

*Corrected and updated as requested*

- Page 7. Note that in the list of taxons in lines 20–21 you combine plant families with plant orders (e.g., Pinales, the only family of this order being Pinaceae).

*Corrected as requested.*

- Page 9 line 21. Check: Crassulaceae instead of Crassulacea. Note that Latin names of families should not be italicized, as a difference with genus and species names.

*Corrected as requested.*

- Page 9, line 17: I think that most of the flavonoids are water-soluble compounds.

*The cited paper by Li et al. reports the following "The major classes of compounds identified from hexane soluble leaf waxes were long chain hydrocarbons, aldehydes, alcohols, esters and β-diketones; flavonoids and triterpenoids."*

- Page 9, Line 25: 2.2.2 Cutin and suberin monomers – Some authors prefer terms such as "structural units" or "units", in the case of macromolecules, and 'monomers' in the case of typical polymers.

*The terms 'cutin and suberin monomer' are frequently and commonly used in contemporary scientific literature in this field. For reasons of clarity and conformity we prefer to stick to such common usage.*

- Page 10, lines 7–12: Some mention could be done to the typical iso- and anteisobranched fatty acids (and alkanes) mainly C15 and C17, for instance Microbiological Reviews (1991), 55, 288–302. Despite these branched chains have also been described in the uropygial gland of birds, they are frequently considered indicators of bacterial metabolism in soil.

*This paragraph deals specifically with cutin and suberin monomers. The suggested additions are outside of this scope.*

- Page 10, line 14 and below: I would improve (clairify) this important section on the effect of temperature: i) specifying better the paragraph about the effect of environmental temperature on lipid-synthesizing organisms and ii) the effect of T on the fate of lipids in the soil. On the other hand, I would discuss more extensively the studies as regards effects of environmental temperature in the degree of saturation of the fatty acids, i.e., the fact that unsaturated FAs increase cell-membrane fluidity, and cell division, favoring the growing of organisms living in very cold environments. Finally, and as indicated above, it would be interesting to explain some effects of extreme temperatures, i.e., the case with forest fires or controlled burning. I think that typical changes in diterpene resin acids and in alkyl series are were first reported in Geoderma, (1988). 42, 115 127.

*This paragraph was updated and a paragraph dealing with the effects of fire was added to a later section (see previous replies to comments).*

- Page 11, line 9: Do not capitalize the name of the species in "Gossypium hirsutum L.".I would not include the initial of Linnaeus due to it is not done in the case of the other species names in the Chapter.

*Changed as requested*

- Page 12, line 14: I would specify e.g., "cauliflower" in addition to "Brassica oleracea" (if this were the case). Depending on the variety, this species include very different plants (subspecies) with very different chemistry, e.g. broccoli, Brussels sprout, etc.

*Varietal was added (gongylodes).*

- Page 12, line 23. Italicize Pinus. Page 12, line 18. I would add "the Poaceae", or "the graminaceous species", or "perennial grass" " before Chionochloa. Page 12, line 25. I would add "the lichen" before Xanthoria.

*Changed as requested*

- Page 15, line 20 and throughout the text: Leave blank space between numbers and units symbols

*Changed as requested*

- Page 17: I think this section is suitable to include some mention to brached fatty acids and alkanes, as the above indicated iso-and anteiso- chains. But also of chlorophyll-derived isoprenoids (phytane, pristane. . .) and other unusual in-chain branched alkenes typical of cyanobacteria.

*We focused on the main compounds. See our previous comments on the scope and page limits.*

- Page 19: I think this section is suitable to include additional information on typical cyclic alkanes indicative of fossil organic matter or contamination with fossil fuels . This would be the case of hopanoids, but also of the classical chromatographic 'hump' of cyclic and branched alkanes generated by e.g., geothermal processes.

*See previous.*

- Page 20, subheading "Transformations and turnover in soil" I think this would be a suitable section to be extended with additional subjects such as the "translocation of lipids" (amongst different soil microcompartments), a typical affect of forest fires, remember the classical studies in Proceedings. Soil Science Society of America (1970) 34:130ăĂ T133. It is also important the "speciation of lipids" in studies on factors involved in soil water repellence (Geoderma (2010) 155: 242–248) or in soils treated with urban wastes, biosolids, or sewage sludges (e.g., Waste Management & Research (2004) 22: 23–24) which systematically lead to a increase of extractable lipids in native soil organic matter, which is progressively incorporated and 'fixed' at different organizational levels of the soil humic colloidal matrix.

*See our previous replies with regards to this section.*

- Page 21, line 1: Amblès

*Corrected as requested*

- Page 21, lines 2–4: Similarly, it is possible to take advantage of the seco- acids as markers of the impact of fire or thermal treatments in soils under pine vegetation, as suggested by classical literature e.g., The Journal of Organic Chemistry 1968., 33: 3718 3722, and Journal of the American Oil Chemists' Society, 1969. 46, 633 634. In

any case, I think that the pioneer booklet by Zinkel et al., (1971) ought to be indicated in the reviews (Diterpene Resin Acids. USDA, Forest Service. Forest Products Laboratory. Madison, Wisconsin. )

*See previous comments*

- Page 22. Differences between different soil compartments. This section would be suitable to introduce some mention to 'lipid speciation' suggested in previous paragraphs. In particular the mechanisms leading to 'fixation' or 'immobilization" of lipids with the humic-type substances are extremely complex and effective, considering that compounds lacking reactive functional groups such as paraffins may be incorporated into condensed forms of organic matter, and require drastic chemical degradation methods for its release (e.g., Soil Biol. Biochem., (1987) 19(5): 513–520). Mechanisms not including encapsulation ("at a molecular level", line 20) but diffusion into microporous structures, or chemisorption in the case of unsaturated chains, fatty acids, etc, may be typical processes. Note that not only "different soil compartments" but also "different plant compartments" are relevant as regards the distribution of lipids. The same alkane or fatty acid molecule may be present in extractive forms as an epicuticular plant wax. Or can be entrapped as nonhydrolyzable ester, or as sterically blocked alkyl chain in cutans, or in complex polyalkyl macromolecules of unknown structure presumptively existing in plants and soils, e.g., Naturwissenschaften (1986) 73: 579–585. In the case of humic-type fractions, it has been indicated that condensation or polymerization of lipid may be an active abiotic mechanism (Naturwissenschaften (1991), 78: 359–362). Nevertheless, the pioneer studies were probably those from Australian soils (Aust. J. Soil Res. (1987) 25;71–82) suggesting a preferential incorporation of alkyl components in soil microaggregates (phenomenon which was also indicated in more recent papers cited in line 30).

*See our previous replies to the general comments.*

- Page 22, line 21. I prefer humic macromolecules or humic substances instead of 'polymers'.

*Changed to organic macromolecules.*

- Page 23, line 1. I prefer alkyl rather than aliphatic (e.g., sugar is aliphatic).

*Aliphatic is a very commonly used term within the context of the molecular proxies used. Seven of the articles sited in our review even use it as a title word.*

- Page 24, line 26. I would change 'biopolymers' (?) by alternative terms such as lipids, biomolecules, homologous series of-, etc (?).

*Changed to 'lipids' as requested.*

- Page 25, lines 1–11: As regards factors determining the presence or the lack of individual molecules in soils, please consider my above suggestions of not overemphasizing selective biodegradation of the lipid compounds, but also immobilization, condensation, insolubilization, fixation or encapsulation as molecules in 'recalcitrant','entrapped', 'fixed', or condensed forms, etc. . .

*See our previous replies.*

- Page 28: General conclusions: I would discuss some of the above processes in the general conclusions

*Transformation processes are only discussed in general terms in the conclusions so we see no room for specifically addressing processes. However, we replaced the specific example of potential disturbance from one dealing with microbial degradation to one dealing with fire induced alterations.*

- Line 20: I would change terms such as 'microbial degradation' by more general ones such as "microbial reworking". This will compensate the (relatively) ill-posed problem about the fate of lipids described in this chapter. Microorganims: i) degrade soil lipids, ii) synthesize alternative lipids which are released to soils, iii) contribute to the biodegradation of the organic matter with an affect of releasing its 'building blocks' and iv) modify the reactivity of organic matter by oxidative processes leading to e.g., increased titrable acidity (both carboxyl and phenolic hydroxyl groups), then humic-type organic matter have increased potential to retain, incorporates or insolubilize lipids into nonextractable forms.

*This issue was solved by replacing the example (see previous reply).*

References
- Check the style of all references and abbreviations of Journals (e.g., page 30, line 20; page 31, line 9).
- Italicize genus (Latin names), as in page 29, lines 23 and 29; page 32, line 12: page 40, line 8; page 42, line 7; page 45, line 2.
- Page 30, line 2:italicize n- in n-alkanes. The same in page 31, line 5; page 32, line 28; page 33, line 9; page 39, line 19; page 43, line 11.
- Page 31, line 6: Do not capitalize unnecessarily "isotopic". The same in some Title words in page 44, line 33.
- Page 34. line 21. Subscript in $CO_2$.
- Page 41, line 34: Spectrom.
- Page 44, line 5: Annu Rev Earth Planetary Sci.

*Changed as requested*

---

## Author Response (AR2)

6 October 2017

**Response to request for technical corrections and title justification "Opportunities and limitations related to the application of plant-derived lipid molecular proxies in soil science" – Boris Jansen & Guido Wiesenberg**

Title

As we also described in our rebuttal to the first round of review, we do not agree with the previous reviewer that our paper exclusively focusses on linear alkyl compounds. We explicitly included cyclic biomarkers such as isoprenoids. However, an important observation of our review is that most of the research related to molecular proxies within soil science has focused on alkyl compounds so far. While we explicitly included cyclic compounds, such as the isoprenoids, in our review, there simply are not as many papers out on those as there are on linear compounds (see also Table 1).

In our previous rebuttal we wrote the following:

"*A clear and motivated delineation of the compound classes considered in our review is given in the introduction (p. 4, l.29 – p. 6, l. 21). These explicitly include the cyclic biomarkers mentioned. It is true that the alkyl compounds receive more attention in the review than some of the other classes of components. This is borne out of necessity, as the majority of research concerning biomarkers in soils focusses on alkyl compounds (see also our reply to the last comment of Referee #1). We explicitly acknowledged the unequal distribution of research efforts with respect to different compound classes on multiple occasions, and urged for future research to help overcome this bias (e.g. p.14, l.24-26; p. 27, l.26-32). With respect to the potential mechanisms of disturbance to consider, clearly a choice had to be made to keep the manuscript within reasonable page limits. We therefore made a motivated choice to focus on three main processes and underpinned this with multiple literature references (p. 5, l. 1-16).*

Because we did explicitly address cyclic biomarkers, we feel it would be imprudent to narrow the title such that this class of components is excluded. Then the title would no longer cover the contents and would erroneously suggest that cyclic biomarkers are not important or were not considered.

Technical corrections

P7 l20-26: request is to write plant names in italics. However, the names presented here are family names and the convention is that family names are not in italics (genus and species are).

P8 l1: request is to change 'fatty acid' to 'n-fatty acid'. However, Mongrand et al. examined not only the saturated, but also the unsaturated fatty acids. Therefore, the prefix 'n' does not apply.

P9 l4: changed as requested

P10 l24-25: request is to change the description of the fatty acids to another convention. However, the convention we used to our knowledge and experience is the most common and established way to annotate such compounds within the field of soil science.

P11 l9: changed as requested

P13 l2: changed as requested

P15 l14-20: changed as requested

P16 l31: changed as requested

P17 l1: changed as requested

P18 l19: changed as requested

P18 l27: changed as requested

P19 l5: changed as requested

P20 l14: changed as requested

P22 l27-30: changed as requested

P23 l16-26: changed as requested

P24 l8-14: changed as requested

P27 l21-22: changed as requested

P28 l21: changed as requested

P29: change hyphen to en-dash in reference page numbers: changed as requested.

P30 l1: changed as requested

P30 l20: changed as requested

P31 l9: changed as requested

P33 l9: changed as requested

P34 l21: changed as requested

P38 l15: genus changed to italics

P39 l9-16: family names should not be in italics

P39 l19: changed as requested

P41 l14: changed as requested

P42 l7: family names should not be in italics

P45 l2: family names should not be in italics

P45 l6: changed as requested

P49: changes to table made as requested